# Online Multi-Armed Bandits with Adaptive Inference

**Maria Dimakopoulou**
Netflix
mdimakopoulou@netflix.com

**Zhimei Ren**
University of Chicago
zmren@statistics.uchicago.edu

**Zhengyuan Zhou**
NYU Stern School of Business
zzhou@stern.nyu.edu

## Abstract

During online decision making in multi-armed bandits, one needs to conduct inference on the true mean reward of each arm based on data collected so far at each step. However, since the arms are adaptively selected–thereby yielding non-i.i.d. data–conducting inference accurately is not straightforward. In particular, sample averaging, which is used in the family of UCB and Thompson sampling (TS) algorithms, does not provide a good choice as it suffers from bias and a lack of good statistical properties (e.g. asymptotic normality). Our thesis in this paper is that more sophisticated inference schemes that take into account the *adaptive* nature of the sequentially collected data can unlock further performance gains, even though both UCB and TS type algorithms are optimal *in the worst case*. In particular, we propose a variant of TS-style algorithms–which we call doubly adaptive TS–that leverages recent advances in causal inference and *adaptively* reweights the terms of a doubly robust estimator on the true mean reward of each arm. Through 20 synthetic domain experiments and a semi-synthetic experiment based on data from an A/B test ran at Netflix, we demonstrate that using an adaptive inferential scheme (while still retaining the exploration efficacy of TS) provides clear benefits in online decision making: the proposed DATS algorithm has superior empirical performance to existing baselines (UCB and TS) in terms of regret and sample complexity in identifying the best arm. In addition, we also provide a finite-time regret bound of doubly adaptive TS that matches (up to log factors) those of UCB and TS algorithms, thereby establishing that its improved practical benefits do *not* come at the expense of worst-case suboptimality.

## 1 Introduction

Stochastic multi-armed bandits [Robbins, 1952, Berry and Fristedt, 1985, Bubeck and Cesa-Bianchi, 2012, Sutton and Barto, 2018, Lattimore and Szepesvári, 2020] is the simplest and most well-established model for sequential decision making, where a decision maker needs to adaptively select an arm at each time from a set of arms with unknown (but fixed) mean rewards, in the hope of maximizing the total accumulated expected returns over a certain time horizon. The existing literature has studied this problem extensively, mainly focusing on developing algorithms that deal with the exploration-exploitation trade-off, yielding at least two broad classes of algorithms that provide optimal (sometimes up to log factors) regret guarantees *in the worst case*.

The first is upper confidence bound (UCB) based algorithms [Lai and Robbins, 1985, Agrawal, 1995, Auer et al., 2002, Auer, 2002, Garivier and Moulines, 2011, Garivier and Cappé, 2011, Carpentier et al., 2011]. Reflecting "optimism in face of uncertainty", UCB algorithms compute confidence bounds of the estimated mean, construct the index for each arm by adding the confidence bound to

35th Conference on Neural Information Processing Systems (NeurIPS 2021).

the mean (as the best statistically plausible mean reward) and select the arm with the highest index. The finite-time regret bounds for these algorithms–$\Theta(\log T)$ gap-dependent bounds and $\Theta(\sqrt{T})$ gap-independent bounds–are both optimal, with the latter in the worst-case sense. On the other hand, UCB algorithms are known to be sensitive to hyper-parameter tuning, and are often hard to tune for stellar performance. This stands in contrast to Thompson sampling (TS) based algorithms [Thompson, 1933, Kaufmann et al., 2012, Ghavamzadeh et al., 2015, Russo et al., 2017], an algorithm that does not require much tuning and that achieves exploration through "probabilistic optimism in face of uncertainty". Further, as a result of this more nuanced exploration scheme, TS has been widely recognized to outperform UCB in empirical applications [Scott, 2010, Graepel et al., 2010, May and Leslie, 2011, Chapelle and Li, 2011]. However, although an old algorithm [Thompson, 1933], its finite-time worst-case regret bound was not known at the time. Consequently, driven by its empirical performance, its theoretical guarantee was raised as an open problem in COLT 2012 [Li and Chapelle, 2012] and was subsequently settled by [Agrawal and Goyal, 2012, 2013c], which yield the same minimax optimal regret guarantees (up to log factors) as in UCB. These developments—both the UCB/TS algorithms and the theoretical guarantees—have subsequently been successfully applied to contextual bandits[1] [Li et al., 2010, Filippi et al., 2010, Chu et al., 2011, Jun et al., 2017, Li et al., 2017, Agrawal and Goyal, 2013a,b, Russo and Van Roy, 2014, 2016, Agrawal et al., 2017].

Despite this remarkably fruitful line of work, searching for better multi-armed bandit algorithms is far from over. For one thing, minimax (i.e., worst-case) optimality is often too conservative a metric, and does not serve as a useful indicator of practical performance for average problems [2]. In particular, an important weakness in UCB/TS algorithms is that the true mean reward of each arm is estimated using a simple sample average, which would work if the data *were* i.i.d. generated. However, the data is adaptively collected and consequently, as pointed out in [Xu et al., 2013, Bowden and Trippa, 2017, Nie et al., 2018, Neel and Roth, 2018, Hadad et al., 2019, Shin et al., 2019] directly using the sample average to estimate the true mean rewards is not guaranteed to be unbiased and asymptotically normal. Intuitively, this is because arms which appear to have worse reward performance than they actually do due to randomness are sampled less often and the downward bias is not corrected. As such, the inferential quality of an estimator–when treating the underlying data *as if* they are i.i.d.–may be poor, thereby constraining the overall online decision making performance.[3]

Our goal in this paper is to leverage recent advances in the causal inference literature that tackle the problem of unbiased and asymptotically normal inference from previously collected, offline bandit data [Hadad et al., 2019, Zhang et al., 2020, Zhan et al., 2021, Bibaut et al., 2021] and obtain practically better online performance by harnessing the strengths of these adaptive inference estimators[4] (originally designed for offline, post-experiment analyses of adaptive experiments) and the effective exploration-exploitation balance provided by TS, thereby designing a more effective *online* sequential decision making algorithm.

## 1.1 Our Contributions

Our contributions are threefold. First, we pinpoint the issues of the existing bandit algorithms (Section 3.1) and design a new algorithm–which we call doubly-adaptive Thompson sampling (DATS)–by incorporating the adaptively-weighted doubly robust estimator in [Hadad et al., 2019] into the online decision making process and making the necessary modifications to the estimated variance of that estimator to render it suitable for the exploration/exploitation trade-off and ensure sufficient exploration. This algorithm mitigates the issues of poor inferential quality inherent in the sample average used in TS/UCB, while retaining the benefits of intelligent exploration (see

---

[1]And to reinforcement learning as well, but contextual bandits and RL are not the focus of this paper.

[2]The fact that TS performs better in practice than UCB, even though the existing UCB bounds are tighter (e.g. by log factors) than those of TS, already attests to that. For instance, Audibert et al. [2009] showed that a variant of UCB has regret $O(\sqrt{KT})$ in multi-armed bandits, while the best known regret bound for TS is $O(\sqrt{KT \log T})$ for TS with Beta prior and $O(\sqrt{KT \log K})$ for TS with Gaussian prior [Agrawal and Goyal, 2017]. And for contextual bandits, TS is often at least a factor of $d$ (context dimension) worse.

[3]Work concurrent to our paper sheds new light on the arm pulling properties of UCB that results in more balanced data collection under small-gap regime compared to TS, which drives the estimates from UCB-collected data towards asymptotic normality in the small-gap case [Kalvit and Zeevi, 2021].

[4]Such as the adaptively weighted estimator proposed by [Luedtke and Van Der Laan, 2016] in the setting of i.i.d. observational data with multiple optimal policies and later generalized by [Hadad et al., 2019] in the multi-armed bandit setting.

Section 3.2 for a more detailed discussion). Previously, inverse propensity weighting (IPW) has been used in bandits[Agarwal et al., 2014], motivated, analyzed and evaluated in [Dimakopoulou et al., 2017, 2019] and systematically benchmarked in [Bietti et al., 2018]. However, IPW has very high variance and these works did not use variance stabilizing weights, which can yield poor inferential performance–as we will see in section 4.2. Second, we validate that the desired benefits in the design of DATS do translate into higher quality decision making. In particular, through 20 synthetic domain experiments and one semi-synthetic experiment, we demonstrate that DATS is more effective than TS and UCB. This effectiveness is shown in two metrics: one is regret, which measures the cumulative performance compared to that of the optimal oracle; the other is sample complexity, which measures how many rounds are needed in order to identify the best arm. Under both metrics, DATS beats all existing bandit algorithms with a clear margin (see Section 4 for a detailed presentation of results). Finally, to complete the picture, we show that DATS achieves this superior practical performance without giving away the comparable worst-case regret guarantees. In particular, we establish that DATS has a finite-time regret of $O\big(K^{3/2}\sqrt{T\log T}\big)$, which is minimax optimal (up to log factors) in the horizon $T$, thus matching that of TS ($K$ is the number of arms). Notably, all existing results on adaptive estimators in the inference literature are asymptotic in nature (typically in the style of central limit theorem bounds), whereas our bound is finite in nature and sheds light in the arena of online decision making, rather than offline inference, on which all prior work on adaptive estimators has focused–to our knowledge. We point out that we do not attempt to be tight in $K$ in our bound, as in the applications we have in mind (e.g. clinical trials, web-service testing), including the one motivating our semi-synthetic experiment, the number of arms is typically a (small) constant. That said, our synthetic empirical results (Fig. 1) on 20 domains (including for large $K$) demonstrate that DATS indeed has an optimal $O(\sqrt{K})$ dependence. We leave tighter analysis on $K$ for future work.

## 2 Problem Formulation

In stochastic multi-armed bandits, there is a finite set $\mathcal{A}$ of arms $a \in \mathcal{A}$ with $|\mathcal{A}| = K$. At every time $t$, the environment generates–in an i.i.d. manner–a reward $r_t(a)$ for every arm $a \in \mathcal{A}$ with expectation $\mathbb{E}[r_t(a)] = \mu_a$, where $\mu_a$ is the unknown true reward of arm $a$ which is modified by mean-zero noise to generate $r_t(a)$. The optimal arm is the arm with the maximum true reward, which is denoted by $a^* := \arg\max_{a \in \mathcal{A}} \mu_a$ and is also unknown. When at time $t$ the decision maker chooses arm $a_t$, *only* the reward of the chosen arm $r_t := r_t(a_t)$ is observed.

At every time $t$, the decision maker employs a policy $\pi_t$ that maps the history of arms and rewards observed up to that time, $\mathcal{H}_{t-1} = (a_0, r_0, \ldots, a_{t-1}, r_{t-1})$, to a probability distribution over the set of arms $\mathcal{A}$ and chooses $a_t \sim \pi_t$. The probability with which arm $a$ is chosen at time $t$ (often referred to as propensity score of arm $a$ at time $t$) is $\pi_{t,a} = \mathbb{P}(a_t = a | \mathcal{H}_{t-1})$. The goal of the decision maker is to make decisions adaptively and learn a sequence of policies $(\pi_1, \ldots, \pi_T)$ over the duration of $T$ time periods, so that the expected cumulative regret with respect to the optimal arm selection strategy over $T$ time periods is minimized, where $\text{Regret}(T, \pi) := \sum_{t=1}^{T} \big(\mu_{a^*} - \mu_{a_t \sim \pi_t}\big)$.

### 2.1 Baseline Algorithms: UCB and TS

Upper confidence bound (UCB) is an algorithm that balances the exploration/exploitation trade-off by forming for each arm $a \in \mathcal{A}$ at every time period $t$ an upper bound $U_{t,a}$ that represents the maximum statistically plausible value of the unknown true reward $\mu_a$ given the history $\mathcal{H}_{t-1}$. Then, at time $t$, UCB chooses the arm with the highest bound, $a_t = \arg\max_{a \in \mathcal{A}} U_{t,a}$. Hence, the policy $\pi_t$ of time $t$ assigns probability one to the arm with the highest bound (breaking ties deterministically) and probability zero to all other arms. An example is UCB-Normal for normally distributed rewards [Auer et al., 2002], where the upper confidence bound of arm $a$ at time $t$ is given by $U_{t,a} = \bar{r}_{t,a} + \beta\sqrt{\hat{\sigma}_{t,a}^2 \log(t-1)}$, where $\bar{r}_{t,a}$ is the sample average and $\hat{\sigma}_{t,a}^2 = \frac{q_{t,a} - n_{t,a}\bar{r}_{t,a}}{n_{t,a}(n_{t,a}-1)}$ is an estimate of the mean reward's variance ($q_{t,a}$ is the sum of squared rewards and $n_{t,a}$ is the number of pulls of arm $a$ up to time $t$) while $\beta$ is an algorithm parameter. Thompson sampling (TS) is another algorithm that balances the exploration/exploitation trade-off by forming for each arm $a \in \mathcal{A}$ at every time period $t$ a posterior distribution $\mathbb{P}(\mu_a \in \cdot | \mathcal{H}_{t-1})$, drawing a sample $\tilde{r}_{t,a}$ from it and choosing the arm with the highest sample, $a_t = \arg\max_{a \in \mathcal{A}} \tilde{r}_{t,a}$. Hence, the policy $\pi_t$ of time $t$ is to choose each arm with the probability that it is optimal given the history of observations $\mathcal{H}_{t-1}$, $\pi_t(a) = \mathbb{P}(a = \text{argmax}_{a \in \mathcal{A}} \tilde{r}_{t,a} | \mathcal{H}_{t-1})$. An example is the TS-Normal, in

which it is assumed that the true reward $\mu_a$ of arm $a$ is drawn from $\mathcal{N}(\hat{\mu}_{0,a}, \hat{\sigma}_{0,a}^2)$, which plays the role of a prior distribution, and the realized reward of arm $a$ at time $t$, $r_t(a)$, is drawn from $\mathcal{N}(\mu_a, \sigma^2)$, where $\mu_a$ is unknown and $\sigma$ is known. The posterior distribution of arm $a$ at time $t$ is also Normal [Russo et al., 2017] with mean $\hat{\mu}_{t,a} = \mathbb{E}[\mu_a | \mathcal{H}_{t-1}] = \frac{\bar{r}_{t,a} \hat{\sigma}_{t-1,a}^2 + \hat{\mu}_{t-1,a} \sigma^2 / n_{t,a}}{\hat{\sigma}_{t-1,a}^2 + \sigma^2 / n_{t,a}}$ and variance $\hat{\sigma}_{t,a}^2 = \mathbb{E}[(\mu_a - \hat{\mu}_{t,a})^2 | \mathcal{H}_{t-1}] = \frac{\hat{\sigma}_{t-1,a}^2 \sigma^2 / n_{t,a}}{\hat{\sigma}_{t-1,a}^2 + \sigma^2 / n_{t,a}}$. The sample average $\bar{r}_{t,a}$ of arm's $a$ observed rewards up to time $t$ plays a prominent role in UCB and TS, since it is used in $U_{t,a}$ and $\mathbb{P}(\mu_a \in \cdot | \mathcal{H}_{t-1})$ respectively. However, the observations of a multi-armed bandit algorithm are adaptively collected and, as a result, they are not i.i.d., which makes these sample averages biased.

## 3 Doubly-Adaptive Thompson Sampling

### 3.1 Motivation

The issue with the sample average (SA) in the posterior $\mathbb{P}(\mu_a \in \cdot | \mathcal{H}_{t-1})$ or in the upper confidence bound $U_{t,a}$ of arm $a$ at time $t$ is that the sample average from adaptively collected data has not being guaranteed to be either unbiased or asymptotically normal [Xu et al., 2013, Bowden and Trippa, 2017, Nie et al., 2018, Hadad et al., 2019, Shin et al., 2019].

$$\bar{r}_{t,a} = \hat{Q}_{t,a}^{\text{SA}} = \frac{\sum_{s=1}^{t} \mathbf{1}(a_s = a) r_s}{\sum_{s=1}^{t} \mathbf{1}(a_s = a)}$$

Intuitively, as explained in [Nie et al., 2018], this general negative bias of the sample average in the adaptive setting stems from the fact that arms which are "unlucky" and their sample average is lower than their true mean are sampled less and this negative bias is not corrected, while arms which are "lucky" and their sample average is higher than their true mean are sampled more and this positive bias is corrected, accounting for a negative bias of the estimator overall. Very recent work of [Kalvit and Zeevi, 2021], which is concurrent to our paper, has shown that the arm-pulling properties of TS is much more unbalanced than those of UCB in the small-gap regime, driving the estimates based on data collected by UCB closer to unbiasedness and asymptotic normality and estimates based on data collected by TS former further away from them. Even though UCB has been recently shown to exhibit this favorable inferential property over TS in the small-gap regime, the widespread empirical superiority of TS over UCB motivates even more towards a greater level of sophistication in the estimator design for TS-based algorithms to correct for the lack of balanced arm-sampling behavior.

Two approaches to correct the bias are the inverse propensity score weighting (IPW) estimator and the doubly-robust (DR) estimator,

$$\hat{Q}_{t,a}^{\text{IPW}} = \frac{1}{t} \sum_{s=1}^{t} \frac{\mathbf{1}(a_s = a)}{\pi_{s,a}} r_s, \quad \hat{Q}_{t,a}^{\text{DR}} = \frac{1}{t} \sum_{s=1}^{t} \left[ \bar{r}_{s-1,a} + \frac{\mathbf{1}(a_s = a)}{\pi_{s,a}} \left( r_s - \bar{r}_{s-1,a} \right) \right]$$

which give an unbiased estimate of arm's $a$ true mean reward $\mu_a$ if the propensity scores $\pi_{s,a}$ are accurate. However, inverse propensity score weighting comes at the expense of high variance–particularly when the propensity scores become small–albeit the variance of DR is smaller than the variance of IPW, since the former uses sample average as the baseline and applies inverse propensity score weighting to a shifted reward using the sample average as a control variate. The other issue of IPW and DR–apart from the high-variance–is that they are not asymptotically normal. Intuitively, this lack of asymptotic normality comes from the fact that in an adaptive setting the importance-sampling ratios converge to 1 for the optimal arm or diverge to infinity for the sub-optimal arms. As a result, both estimators' variance may either be dominated by their first terms or their last terms depending on the arm. At a more theoretical level, in an adaptive setting this violates the classical condition of martingale central limit theorems that the conditional variance of the terms given previous observations stabilizes asymptotically [Hall and Heyde, 2014][5].

An approach that has been proposed in recent literature for performing offline inference from previously collected adaptive data is to use adaptive weights, which are weights that are a function of the history. [Luedtke and Van Der Laan, 2016] first proposed adaptive weights for estimating

---

[5]Section B.1. of the Supplemental Material visualizes the shortcomings of sample average, IPW and DR in a toy domain of adaptive data collection introduced in [Hadad et al., 2019].

the expected reward under the optimal policy when one has access to i.i.d. observational data and multiple optimal policies exist. Subsequently, [Hadad et al., 2019] developed the adaptively weighted method for offline inference on adaptive data, such as the data collected by a multi-armed bandit and produced the unbiased and asymptotically normal adaptive doubly-robust estimator (ADR). The approach in [Hadad et al., 2019] modifies the DR estimator by weighing the efficient score

$$\hat{\Gamma}_{s,a} := \bar{r}_{s-1,a} + \frac{\mathbf{1}(a_s = a)}{\pi_{s,a}} \left( r_s - \bar{r}_{s-1,a} \right)$$

of each data point $s \in [t] := \{1, \ldots, t\}$ by a non-uniform weight $w_{s,a}$ instead of weighing all data points uniformly with weight $1/t$. The resulting estimator is a weighted average of the efficient scores, which is referred to as adaptive doubly-robust (ADR), since these weights $w_{s,a}$ are *adapted* to the history $\mathcal{H}_{s-1}$. ADR takes the form

$$\hat{Q}_{t,a}^{\text{ADR}} = \frac{\sum_{s=1}^{t} w_{s,a} \hat{\Gamma}_{s,a}}{\sum_{s=1}^{t} w_{s,a}}$$

The intuition behind ADR is that if the contribution of each term $\Gamma_{s,a}$ to the variance of the DR estimator is unequal, then using uniform weights is not as efficient (i.e., results in larger variance) as using weights inversely proportional to the standard deviation of each term $\Gamma_{s,a}$. Since the data collection at time $s$ adapts to the history $\mathcal{H}_{s-1}$ via the propensity score $\pi_{s,a}$, so does the variance of $\Gamma_{s,a}$. As a result, the chosen weight $w_{s,a}$ is also *adaptive* to the history $\mathcal{H}_{s-1}$. In the i.i.d. setting with multiple optimal policies, [Luedtke and Van Der Laan, 2016] proposed weighing $\Gamma_{s,a}$ by $w_{s,a} = \frac{\sqrt{\pi_{s,a}/t}}{\sum_{s'=1}^{t} \sqrt{\pi_{s',a}/t}}$. Subsequently, [Hadad et al., 2019], proposed a generalized mechanism for constructing weights $w_{s,a}$ in an adaptive setting such that the conditions of infinite sampling, variance convergence and bounded moments are satisfied–which are in turn necessary for the resulting ADR estimator to be unbiased, have low variance and an asymptotically normal distribution–and, among others, retrieved the weighting scheme proposed in [Luedtke and Van Der Laan, 2016].

## 3.2  Algorithm

We propose *doubly-adaptive Thompson sampling* (DATS), which uses as building block

$$\hat{Q}_{t,a}^{\text{ADR}} = \frac{\sum_{s=1}^{t} \sqrt{\pi_{s,a}} \hat{\Gamma}_{s,a}}{\sum_{s=1}^{t} \sqrt{\pi_{s,a}}}$$

In online learning, the decision maker knows *exactly* the propensity scores in $\hat{Q}_{t,a}^{\text{ADR}}$. Additionally, in TS-based algorithms, these propensity scores are non-trivial and are bounded away from zero and one–at least in the initial stages of learning–unlike UCB, in which the propensity score is equal to one for the chosen arm and equal to zero for all other arms. With TS, at any time $t$ the propensity scores $\pi_{t,a}$ can be computed analytically or via Monte Carlo sampling based on the sampling distribution of each arm $a$ and then logged in order to be used for inference in subsequent time steps rather than the need to fit a propensity model in the UCB setting, which may not be accurate. For this reason, our new multi-armed bandit algorithm belongs to the TS class rather than the UCB class.

At each time step $t \in [T]$, where $T$ is the learning horizon, DATS forms a reward sampling distribution for each arm $a \in \mathcal{A}$ based on the history of observations $\mathcal{H}_{t-1}$ collected so far. The reward sampling distribution of arm $a$ at time $t$ is chosen to be normal $\mathcal{N}(\hat{\mu}_{t,a}, \hat{\sigma}_{t,a}^2)$ with mean and variance

$$\hat{\mu}_{t,a} := \hat{Q}_{t,a}^{\text{ADR}} = \frac{\sum_{s=1}^{t} \sqrt{\pi_{s,a}} \hat{\Gamma}_{s,a}}{\sum_{s=1}^{t} \sqrt{\pi_{s,a}}}, \quad \hat{\sigma}_{t,a}^2 := \frac{\sum_{s=1}^{t} \pi_{s,a}[(\hat{\Gamma}_{s,a} - \hat{\mu}_{t,a})^2 + 1]}{\left( \sum_{s=1}^{t} \sqrt{\pi_{s,a}} \right)^2}$$

Note, that the variance in the sampling distribution of DATS has an auxiliary '+1' term compared to the variance in [Hadad et al., 2019]. This is essentially important for the online setting, as the auxiliary term guarantees a lower bound of the estimated variance, thereby ensuring sufficient exploration in the initial stages of learning (in contrast to the offline setting in Hadad et al. [2019]). The derivation of $\hat{\sigma}_{t,a}$ used by DATS can be found in the Supplemental Material section A.1. At time $t$, the sampling distributions of all arms are used to compute the probability of arm $a$ being chosen, i.e., its propensity score. Given that TS is a probability matching heuristic, i.e., it plays an arm with the probability

---
**Algorithm 1** Doubly-Adaptive Thompson Sampling
---

**Input:** uniform exploration parameter $\gamma \in (0,1)$ over non-eliminated arms; sampling parameter $\kappa$;

**for** $a \in \mathcal{A}$ **do**
  Pull arm $a$ and observe reward $r_{\text{initial},a}$.
  Initialize $\bar{r}_{0,a} \leftarrow r_{\text{initial},a}$, $n_{0,a} \leftarrow 1, \pi_{1,a} \leftarrow 1/K$.
**end for**
$\mathcal{A}_1 \leftarrow \mathcal{A}$
**for** $t = 1$ **to** $T$ **do**
  Pull arm $a_t \sim \text{Multinomial}\left(\mathcal{A}_t, (\pi_{t,a})_{a \in \mathcal{A}_t}\right)$ and observe reward $r_t \leftarrow r_t(a_t)$.
  ▷ Update sample averages and counts.
  $\bar{r}_{t,a_t} \leftarrow \frac{n_{t-1,a_t}\bar{r}_{t-1,a_t}+r_t}{n_{t-1,a_t}+1}, n_{t,a_t} \leftarrow n_{t-1,a_t} + 1$ and $\bar{r}_{t,a} \leftarrow \bar{r}_{t-1,a}, n_{t,a} \leftarrow n_{t-1,a} \, \forall a \neq a_t$
  ▷ Update sampling distributions.
  **for** $a \in \mathcal{A}_t$ **do**
    $\hat{\Gamma}_{s,a} \leftarrow \bar{r}_{s-1,a} + \mathbf{1}\{a_s = a\}\frac{r_s - \bar{r}_{s-1,a}}{\pi_{s,a}}, \forall s \in [t],$
    $\hat{\mu}_{t,a} \leftarrow \frac{\sum_{s=1}^t \sqrt{\pi_{s,a}}\hat{\Gamma}_{s,a}}{\sum_{s=1}^t \sqrt{\pi_{s,a}}}, \quad \hat{\sigma}_{t,a}^2 \leftarrow \frac{\sum_{s=1}^t \pi_{s,a}\left[(\hat{\Gamma}_{s,a}-\hat{\mu}_{t,a})^2+1\right]}{\left(\sum_{s=1}^t \sqrt{\pi_{s,a}}\right)^2}$
  **end for**
  ▷ Perform arm elimination.
  $p_{t+1,a} \leftarrow \min_{a' \in \mathcal{A}_t} \Phi\left((\hat{\mu}_{t,a} - \hat{\mu}_{t,a'})/\kappa\sqrt{\hat{\sigma}_{t,a}^2 + \hat{\sigma}_{t,a'}^2}\right) \forall a \in \mathcal{A}_t$
  $\mathcal{A}_{t+1} \leftarrow \mathcal{A}_t - \{a \in \mathcal{A}_t : p_{t+1,a} < 1/T\}$
  ▷ Compute propensity scores.
  $\tilde{r}_{t+1,a} \sim \mathcal{N}(\hat{\mu}_{t,a}, \kappa^2\hat{\sigma}_{t,a}^2)$ and $\pi_{t+1,a} \leftarrow \mathbb{P}\left(a = \text{argmax}_{a' \in \mathcal{A}_{t+1}} \tilde{r}_{t+1,a} \mid \mathcal{H}_t\right), \forall a \in \mathcal{A}_{t+1}$
  $\pi_{t+1,a} \leftarrow (1-\gamma)\pi_{t+1,a} + \gamma/|\mathcal{A}_{t+1}|, \forall a \in \mathcal{A}_{t+1}$
**end for**
---

that it is optimal according to the posteriors, the propensity score of arm $a$ at time $t$ is equal to the probability with which a single sample $\tilde{r}_{t,a} \sim \mathcal{N}(\hat{\mu}_{t,a}, \kappa^2\hat{\sigma}_{t,a}^2)$ from the sampling distribution of arm $a$ is greater than a single sample $\tilde{r}_{t,a'} \sim \mathcal{N}(\hat{\mu}_{t,a'}, \kappa^2\hat{\sigma}_{t,a'}^2)$ from the sampling distribution of any other arm $a'$. Hence, $\pi_{t,a} = \mathbb{P}(a = \text{argmax}_{a' \in \mathcal{A}_t} \tilde{r}_{t,a'} \mid \mathcal{H}_{t-1})$, where $\tilde{r}_{t,a'} \sim \mathcal{N}(\hat{\mu}_{t,a'}, \kappa^2\hat{\sigma}_{t,a'}^2)$. These propensity scores can be computed accurately via Monte Carlo simulation.

In order to control the variance of their estimator, [Hadad et al., 2019] derived the asymptotic normality of their $\hat{Q}^{\text{ADR}}$ under the assumption that each arm has a non-negligible probability of being chosen, i.e., the propensity score of arm $a$ at every time $t$ is lower bounded—this however cannot be assumed in the online setting since otherwise the sub-optimal arms will be pulled for a non-trivial fraction of times and yield undesired regret. In order to deal with diminishing propensity scores in the online setting and derive a regret upper bound, DATS performs an arm elimination step. To do this, we compute the probability that a sample from the posterior of arm $a$ is greater than a sample from the posterior of arm $a'$. The difference $D_{a,a',t} = \tilde{r}_{a,t} - \tilde{r}_{a',t}$ is normally distributed $\sim \mathcal{N}(\hat{\mu}_{a,t} - \hat{\mu}_{a',t}, \kappa^2(\hat{\sigma}_{a,t}^2 + \hat{\sigma}_{a',t}^2))$. So, $P(D_{a,a',t} > 0) = \Phi\left((\hat{\mu}_{t,a} - \hat{\mu}_{t,a'})/\kappa\sqrt{\hat{\sigma}_{t,a}^2 + \hat{\sigma}_{t,a'}^2}\right)$, where $\Phi$ denotes the CDF of a standard normal distribution. If there exists an arm $a'$ for which $P(D_{a,a',t} > 0) < 1/T$ (i.e., arm $a$ is dominated by $a'$), then arm $a$ is eliminated. The set of arms available to the decision maker at the beginning of time step $t + 1$ is denoted by $\mathcal{A}_{t+1}$. Also, DATS maintains a level of uniform exploration $\gamma$ among the non-eliminated arms, where $\gamma$ is a parameter of the algorithm (default $\gamma = 0.01$) and controls how small propensity scores get[6]. Our default choice for the sampling parameter $\kappa$ is 1.

---

[6]Note that the presence of this uniform exploration does *not* yield linear regret, due to the existence of the arm elimination step in our algorithm and the fact that the algorithm applies uniform exploration of magnitude $\gamma$ *only* over the set of the non-eliminated arms. Indeed, in section 5, we prove that the regret of DATS is $O\left(K^{3/2}\sqrt{T \log T}\right)$, which is minimax optimal (up to log factors) in the horizon $T$

# 4 Empirical Evaluation

We now present computational results that demonstrate the robustness of DATS in comparison to baselines, both in terms of regret and in terms of sample complexity for identifying the best arm.

## 4.1 Synthetic Experiments

First, we present results on synthetic domains with varying number of arms and noise levels, as commonly done in literature [Russo, 2016]. We simulate 20 synthetic domains with $K = 5, 10, 15, 20, 25, 30, 35, 40, 45, 50$ where the true mean reward $\mu_a$ of each arm $a$ is drawn randomly from $\mathcal{N}(0, 0.125)$ in the low signal-to-noise ratio (SNR) setting and from $\mathcal{N}(0, 0.5)$ in the high SNR setting, while the rewards $r_{t,a}$ are drawn from $\mathcal{N}(\mu_a, 1)$. The multi-armed bandits run for horizon $T = 10000$. For TS-Normal [Lattimore and Szepesvári, 2020], described in section 2.1, we select a weak prior $\hat{\mu}_{0,a} = 0$, $\hat{\sigma}_{0,a}^2 = 10^6$ for all arms $a \in \mathcal{A}$ and known noise variance $\sigma = 1$. For UCB-Normal [Auer et al., 2002], described in section 2.1, we tune $\beta$ among values $1, 1.5, 2, 2.5, 3, 4$ and select the one with the best performance in each domain. For DATS, we select $\gamma = 0.01$, $\kappa = 1$. Finally, we also evaluate a simplified version of DATS, called DATS-clipping: instead of arm elimination and uniform exploration among the non-eliminated arms that DATS introduces to control diminishing propensity scores, DATS-clipping applies TS to a modified ADR estimator which replaces $\pi_{s,a}$ in $\Gamma_{s,a}$ and in the adaptive weights $w_{s,a}$ by the clipped propensity $\max(\delta, \pi_{s,a})$. Propensity score clipping is a common practice in the offline evaluation literature [Crump et al., 2009], but this modified estimator is no-longer unbiased. DATS-clipping uses $\delta = 0.001$, $\kappa = 1$.

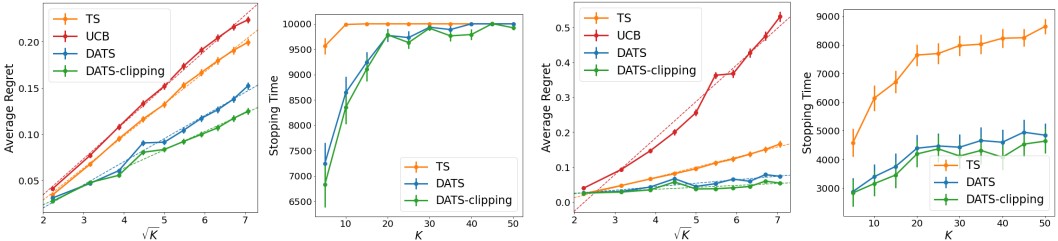

(a) Average regret and stopping power in low SNR.     (b) Average regret and stopping power in high SNR.

Figure 1: Comparison of TS, UCB, DATS and DATS-clipping in terms of average regret vs. $\sqrt{K}$ and stopping power vs. $K$ (for probabilistic algorithms) in 20 synthetic domains (10 arm settings $K = 5, ..., 50$ and 2 noise settings). Each domain is simulated 64 times and 1 standard error is shown.

We compare TS, UCB, DATS and DATS-clipping in terms of average regret, $\frac{1}{T}\sum_{t=1}^{T}(\mu_{a^*} - \mu_{a_t})$, as a function of the square root number of arms, $\sqrt{K}$, shown in the left subplots of 1a and 1b for the low SNR and the high SNR setting respectively. We also compare the probabilistic bandits–TS, DATS, DATS-clipping–in terms of sample complexity for identifying the best arm with error tolerance $\epsilon = 0.05$ (i.e., with 95% confidence) using the Bayesian stopping criterion, which stops at the first time $t^\dagger$ when there is an arm $a^\dagger$ such that its propensity score $\pi_{t^\dagger,a^\dagger}$ computed from its posterior $\mathbb{P}(\mu_{a^\dagger} \in \cdot|\mathcal{H}_{t^\dagger-1})$ satisfies $\pi_{t^\dagger,a^\dagger} \geq 1-\epsilon$ [Russo, 2016]. The right subplots of of 1a and 1b show this stopping time $t^\dagger$ for each algorithm as a function of the number of arms $K$ for the low SNR and the high SNR setting respectively. Each one of the 20 domains is simulated over 64 simulations and the plots in Figure 1 show the mean and one standard error for both the average regret and the stopping power metric. Both DATS and the DATS-clipping heuristic attain significantly better performance both in terms of average regret (linear dependence on $\sqrt{K}$ with smaller slope than TS and UCB) and in terms of stopping power (identify the best arm with 95% confidence); DATS and DATS-clipping have better sample complexity than TS in all high SNR domains and in the low SNR domains with small $K$, while being within error from one-another.

## 4.2 Semi-Synthetic Experiment Based on A/B Test Data

We now evaluate the performance of DATS compared to baselines using data from an A/B test with 6 user-interface (UI) test cells (1 control/production cell and 5 new treatment cells) ran at Netflix.

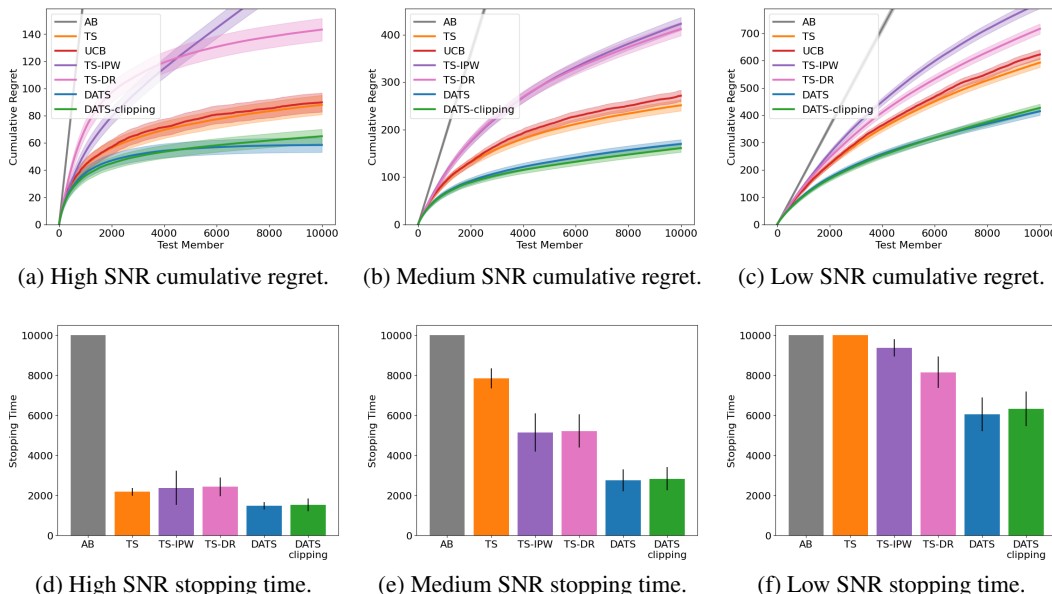

<p align="center">(a) High SNR cumulative regret.    (b) Medium SNR cumulative regret.    (c) Low SNR cumulative regret.</p>

<p align="center">(d) High SNR stopping time.    (e) Medium SNR stopping time.    (f) Low SNR stopping time.</p>

Figure 2: Comparison of A/B test, TS, UCB, TS-IPW, TS-DR, DATS and DATS-clipping in terms of cumulative regret and 95% confidence stopping time (for stochastic algorithms) in high SNR ($\sigma = 0.32$), medium SNR ($\sigma = 0.64$) and low SNR ($\sigma = 1.28$) with $K = 6$ arms. The true reward means and medium SNR $\sigma$ estimated by Netflix's A/B test data. Bandits run for horizon $T = 10000$. Results are averaged over 64 simulations and 95% confidence intervals are shown.

For each test member that took part in this A/B test, one of the 6 UI cells was selected uniformly at random an was presented to the member throughout the duration of the A/B test. Measures of member satisfaction were collected throughout the A/B test, from which a target reward was defined. From the member-level data collected by the end of the A/B test, we extracted the sample mean of the target reward corresponding to each UI cell relative to the control/production UI cell, $\mu = [0, -0.05, 0.15, 0.02, 0.28, 0.2]$, and the sample standard deviation of that target reward across cells, $\sigma = 0.64$. For reproducibility purposes, we used the above $\mu$ and $\sigma$ to simulate a multi-armed bandit domain with $K = 6$ arms and $T = 10000$ observations, where rewards are generated from $\mathcal{N}(\mu, \sigma^2 I)$ (I is the $6 \times 6$ identity matrix).[7]. We simulate 3 noise domains: medium SNR with $\sigma = 0.64$ (observed in the A/B test); high SNR with $\sigma = 0.32$ (halving the observed noise standard deviation) and low SNR with $\sigma = 1.28$ (doubling the observed noise standard deviation).

We evaluate TS ($\hat{\mu}_{0,a} = 0$, $\hat{\sigma}^2_{0,a} = 10^6$, oracle $\sigma$), UCB ($\beta = 1, 1.5, 2, 2.5, 3, 4$), DATS ($\gamma = 0.01$, $\kappa = 1$) and DATS-clipping ($\delta = 0.001$, $\kappa = 1$), as in section 4.1. Additionally, we benchmark DATS against two more TS-based algorithms–TS-IPW and TS-DR–that replace the biased sample mean with the unbiased but not adaptively-weighted estimators $\hat{Q}^{\text{IPW}}$ and $\hat{Q}^{\text{DR}}$ respectively (more details in section 3.1) instead of $\hat{Q}^{\text{ADR}}$ as in DATS. The rest of the TS-IPW and TS-DR is implemented as in DATS (Algorithm 1) with $\gamma = 0.01$, $\kappa = 1$[8].

Figure 2 shows that DATS and DATS-clipping have a clear advantage over baselines both in terms of cumulative regret and in terms of sample complexity for identifying the best arm with 95% confidence in all three noise settings. Interestingly, TS-IPW and TS-DR, although unbiased, have worse regret performance than both TS and UCB, which use the biased sample mean. TS and best-tuned UCB are within error from each other. This indicates that using estimators which correct the sample mean bias stemming from the adaptive data collection is not enough if special attention is not given to the variance properties of these estimators. Both TS-IPW and TS-DR suffer from high variance due to the

---

[7]The true mean reward $\mu_a$ of each arm $a$ and the noise $\sigma$ in this experiment are grounded in real data, hence its characterization as semi-synthetic (unlike the Section 4.1 where $\mu_a$ is generated at random for each arm $a$.)

[8]Section B.2 of the Supplemental Material explores tuning parameter $\gamma$ controlling the uniform exploration over non-eliminated arms in TS-IPW, TS-DR, DATS and parameter $\delta$ clipping the propensities in DATS-clipping.

propensity scores in the denominator of their estimator which become small in an online multi-armed bandit setting[9]. Through the variance-stabilizing property of the adaptive weights, DATS is unbiased with reasonably controlled variance and is able to provide a better confidence region for all arms' values, thus improving learning in the online setting.

## 5 Finite-Time Regret Guarantees

Our main theoretical result concerns an upper bound of the finite-time regret resulting from Algorithm 1. We present the proof of the two-arm setting in this section, whereas the generalization to the $K$-arm case is deferred to the supplemental material. Theorem 1 formally shows that for the two-arm stochastic bandit problem the expected regret resulting from Algorithm 1 matches the minimax lower bound $\Omega(\sqrt{T})$ (up to logarithmic factors). We also show a problem-dependent bound $O(\log T/\Delta)$ for our algorithm. Throughout, we assume that the standard deviation of the noise is $\sigma$, and that there exists $M > 0$ such that $|r_t(a)| \leq M$ for $t \in [T]$ and $a \in \mathcal{A}$.

**Theorem 1** *For the two-arm stochastic bandit problem, the expected regret incurred by Algorithm 1 is bounded as* $\mathbb{E}[R_T] \leq O\big(\sqrt{T \log T}\big)$.

Without loss of generality, let $\mathcal{A} = \{1, 2\}$ and arm 1 be the optimal arm; let $\Delta = \mu_1 - \mu_2$ denote the gap between the two arms. The proof of Theorem 1 starts from establishing that the estimator $\hat{\mu}_{t,a}$ concentrates around the true mean $\mu_a$ with high probability; since $\hat{\mu}_{t,a}$ is a good estimator for $\mu_a$, we claim that we are able to recognize the optimal arm fast enough—so that we do not pay too much in the exploration phase—and we are committed to the optimal arm afterwards.

To start, we define $M_{t,a} := \sum_{s=1}^{t} \sqrt{\pi_{s,a}} \cdot \big(\hat{\Gamma}_{s,a} - \mu_a\big)$. By construction, $\{M_{s,a}\}_{s \geq 1}$ is a martingale w.r.t. the filtration $\{\mathcal{H}_s\}_{s \geq 1}$. Let $\tau(a) := \min\{t : p_{t+1,a} < 1/T\}$. Since $p_{t+1,a}$ is measurable w.r.t. $\mathcal{H}_t$ for any $t \in [T]$, $\tau(a)$ is a stopping time w.r.t. the filtration $\{\mathcal{H}_t\}_{t \geq 1}$. Consequently, $S_{t,a} := M_{\tau(a) \wedge t,a}$ is a (stopped) martingale w.r.t. $\{\mathcal{H}_t\}_{t \geq 1}$. For any $a \in \mathcal{A}$, $t \in [T]$, the martingale difference of $S_{t,a}$ is given by

$$D_{t,a} = S_{t,a} - S_{t-1,a} = M_{\tau(a) \wedge t,a} - M_{\tau(a) \wedge (t-1),a} = \mathbf{1}\{\tau(a) \geq t\} \cdot \sqrt{\pi_{t,a}} \cdot \big(\hat{\Gamma}_{t,a} - \mu_a\big).$$

On the event $\{\tau(a) \geq t\}$, the absolute value of the martingale difference $|D_{t,a}|$ can be bounded as,

$$\sqrt{\pi_{t,a}} \cdot |\hat{\Gamma}_{t,a} - \mu_a| = \sqrt{\pi_{t,a}} \cdot \left| \bar{r}_{t-1,a} - \mu_a + \frac{\mathbf{1}\{A_t = a\}}{\pi_{t,a}} \cdot (r_t - \bar{r}_{t-1,a}) \right| \leq 2M + \frac{2M}{\sqrt{\gamma}}.$$

Consequently, $\sum_{s=1}^{t} |D_{s,a}|^2 \leq 4M^2 \cdot (1 + 1/\sqrt{\gamma})^2 \cdot t$. On the other hand,

$$\sum_{s=1}^{t} \mathbb{E}\big(D_{s,a}^2 \mid \mathcal{H}_{s-1}\big) = \sum_{s=1}^{t} \mathbf{1}\{\tau(a) \geq s\} \cdot \big[(1 - \pi_{s,a}) \cdot (\bar{r}_{s,a} - \mu_a)^2 + \sigma^2\big] \leq (4M^2 + \sigma^2) \cdot t.$$

We then make use of the following lemma to establish the concentration result.

**Lemma 1 (Theorem 3.26 of Bercu et al. [2015])** *Let $\{M_n\}_{n \geq 1}$ be a square integrable martingale w.r.t. the filtration $\{\mathcal{F}_n\}_{n \geq 1}$ such that $M_0 = 0$. Then, for any positive $x$ and $y$,*

$$\mathbb{P}\left( M_n \geq x, \ \sum_{i=1}^{n} (M_i - M_{i-1})^2 + \sum_{i=1}^{n} \mathbb{E}\big[(M_i - M_{i-1})^2 \mid \mathcal{F}_{i-1}\big] \leq y \right) \leq \exp\left( -\frac{x^2}{2y} \right).$$

Applying Lemma 1 to $S_{t,a}$, we have for any $x > 0$,

$$\mathbb{P}\left( S_{t,a} \geq x \right) = \mathbb{P}\left( S_{t,a} \geq x, \sum_{s=1}^{t} D_{s,a}^2 + \sum_{s=1}^{t} \mathbb{E}[D_{s,a}^2 \mid \mathcal{H}_{s-1}] \leq \left( 4M^2 \cdot \big(2 + 2/\sqrt{\gamma} + 1/\gamma\big) + \sigma^2 \right) \cdot t \right)$$

$$\leq \exp\left( -\frac{x^2}{\big(8M^2 \cdot (2 + 2/\sqrt{\gamma} + 1/\gamma) + 2\sigma^2\big) \cdot t} \right).$$

We now consider a "good" $E$ event on which the optimal arm is identified within $T_0 = O\big(\log(T)/\Delta^2\big)$ pulls, i.e., $E := \{\tau(2) \leq T_0\}$. Lemma 2, whose proof is deferred to Supplementary Section A.1, guarantees the good event happening with high probability.

---

[9]TS-DR has better variance properties than TS-IPW, and thus a bit better performance in the online setting.

**Lemma 2** *The event $E$ happens with probability at least $1 - 2/T^2 - 2(\log T)^3/T^2$.*

On the good event $E$, the sub-optimal arm is pulled for at most $T_0$ times, thus incurring a regret at most $T_0 \cdot \Delta$. We can decompose the expected regret as,

$$\mathbb{E}\big[\text{Regret}(T,\pi)\big] = \mathbb{E}\big[\text{Regret}(T,\pi) \cdot \mathbf{1}\{E\}\big] + \mathbb{E}\big[\text{Regret}(T,\pi) \cdot \mathbf{1}\{E^c\}\big]$$
$$\leq T_0\Delta + TM \cdot \frac{2}{T^2} + TM \cdot \frac{2(\log T)^3}{T^2} = c(M,\gamma) \cdot \frac{\log(T)}{\Delta} \qquad (1)$$

where $c(M,\gamma)$ is a constant that only depends on $M$ and $\gamma$. (1) provides a problem-dependent bound. To see the problem-independent bound, note that

$$\mathbb{E}\big[\text{Regret}(T,\pi)\big] = \mathbf{1}\Big\{\Delta < \frac{\sqrt{c(M,\gamma)\log(T)}}{\sqrt{T}}\Big\} \cdot \mathbb{E}\big[\text{Regret}(T,\pi)\big]$$
$$+ \mathbf{1}\Big\{\Delta \geq \frac{\sqrt{c(M,\gamma)\log(T)}}{\sqrt{T}}\Big\}\mathbb{E}\big[\text{Regret}(T,\pi)\big] \leq \sqrt{c'(M,\gamma)T\log(T)}.$$

As a by-product of the proof of Theorem 1, we obtain a high probability regret bound, which states that with probability at least $1 - 2/T^2 - 2 \cdot (\log T)^3/T^2$, the regret can be bounded as $\text{Regret}(T,\pi) = O\big(\sqrt{T \log T}\big)$.

Finally, we state the result for the general $K$-arm case, where we consider the case that for each arm $a$, the suboptimality gap $\Delta_a \leq M\sqrt{K}$. In our proof, we consider a slightly modified version of the algorithm where we take the sampling parameter to be $\kappa_1 = \frac{16MK}{\gamma}$ for $t < 8(1 + \sqrt{K/\gamma})^2 \cdot \log T$, and $\kappa_2 = \frac{16\sqrt{K}M}{\sqrt{\gamma}}$ otherwise; we clip the estimated variance $\hat{\sigma}_{t,a}$ by $c_{\max}\sqrt{t}/(\sum_{s=1}^{t}\sqrt{\pi_{s,a}})$, where $c_{\max}$ can be a large constant. The proof of the result can be found in Supplementary Section A.2.

**Theorem 2** *For the $K$-arm stochastic bandit problem, the expected regret incurred by Algorithm 1 is bounded as $\max_{\Delta_a \leq M\sqrt{K}} \mathbb{E}\big[\text{Regret}(T,\pi)\big] \leq O\big(K^{3/2} \cdot \sqrt{T\log T}\big)$.*

## 6 Conclusions & Societal Impact

Experimentation consumes resources and hence is expensive in the real-world. As such, experimentation methods that achieve better data efficiency would be desirable for our increasingly AI-powered society. Adaptive experimentation holds great promise over traditional A/B tests for real-world domains (e.g. web-service testing, clinical trials etc.), as it aims to maximize the test population's reward during learning and it is effective in reducing the test duration by directing test-traffic into actions that are more promising rather than splitting traffic equally among arms. In this work, we advance the field of adaptive experimentation by introducing the idea of reliable inference *during* online decision making. The adaptive data collection in an online multi-armed bandit setting poses an inference challenge that is overlooked in traditional, provably optimal and widely adopted algorithms such as UCB and TS. We design an online multi-armed bandit algorithm, called *doubly-adaptive Thompson sampling (DATS)*, which incorporates advances in offline causal inference estimators for adaptive experimental designs and modifies them appropriately in order to render them suitable for the exploration/exploitation trade-off present in online multi-armed bandits. DATS is shown empirically to improve the test population reward and decrease the test samples required to identify the best arm, while attaining the optimal worst-case regret bound guarantees in terms of time horizon (proven theoretically) and in terms of number of arms (shown empirically). Beyond DATS, the aim for this paper is to motivate a general approach for designing online learning algorithms with more accurate estimation methods that accounts for the biases stemming from the data-collecting mechanism.

## 7 Acknowledgments

Zhengyuan Zhou would like to gratefully acknowledge the generous support from the NYU Stern Fall 2021 Center for Global Economy and Business research grant and JP Morgan AI research grant, which played a major role in enabling the rapid development of the results in this paper.

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
