Supplemental Material for:

# Online Multi-Armed Bandits with Adaptive Inference

Maria Dimakopoulou, Zhimei Ren, Zhengyuan Zhou

## A   Supplementary Proofs

### A.1   Proof of Lemma 2

To start, we define

$$T_0 = \frac{1}{\gamma\Delta^2} \cdot \left( \kappa\sqrt{8M^2/\gamma + 2} \cdot \sqrt{2\log T - \log\log T} + 8M\sqrt{3 + 2/\sqrt{\gamma} + 1/\gamma)} \cdot \sqrt{\log T} \right)^2,$$

where $\kappa > 0$ is a constant to be specified later. The probability of the good event $E$ not happening can be decomposed as,

$$\mathbb{P}(E^c) = \mathbb{P}(\tau(2) > T_0) = \mathbb{P}(\tau(1) \le T_0, \tau(2) > T_0) + \mathbb{P}(\tau(1) > T_0, \tau(2) > T_0).$$

The first term corresponds to the probability of eliminating the optimal arm within the first $T_0$ pulls:

$$\mathbb{P}(\tau(1) \le T_0, \tau(2) > T_0) = \sum_{t=1}^{T_0} \mathbb{P}(\tau(1) = t, \tau(2) > T_0) \tag{2}$$

Note that

$$\tau(1) = t \implies \mathbb{P}(\tilde{r}_{t+1,1} > \tilde{r}_{t+1,2} \mid \mathcal{H}_t) < \frac{1}{T}$$

$$\Longleftrightarrow \mathbb{P}\left( \frac{\tilde{r}_{t+1,1} - \hat{\mu}_{t,1} - \tilde{r}_{t+1,2} + \hat{\mu}_{t,2}}{\kappa\sqrt{\hat{\sigma}_{t,1}^2 + \hat{\sigma}_{t,2}^2}} > \frac{\hat{\mu}_{t,2} - \hat{\mu}_{t,1}}{\kappa\sqrt{\hat{\sigma}_{t,1}^2 + \hat{\sigma}_{t,2}^2}} \,\Big|\, \mathcal{H}_t \right) < \frac{1}{T}$$

$$\Longleftrightarrow 1 - \Phi\left( \frac{\hat{\mu}_{t,2} - \hat{\mu}_{t,1}}{\kappa\sqrt{\hat{\sigma}_{t,1}^2 + \hat{\sigma}_{t,2}^2}} \right) < \frac{1}{T}$$

$$\implies \frac{\hat{\mu}_{t,2} - \hat{\mu}_{t,1}}{\kappa\sqrt{\hat{\sigma}_{t,1}^2 + \hat{\sigma}_{t,2}^2}} > \sqrt{2\log T - 2\log\log T}.$$

Consequently, we can upper bound (2) as follows:

$$(2) \le \sum_{t=1}^{T_0} \mathbb{P}\left( \tau(1) = t, \tau(2) > T_0, \hat{\mu}_{t,2} - \hat{\mu}_{t,1} > \kappa \cdot \sqrt{2\log T - 2\log\log T} \cdot \sqrt{\hat{\sigma}_{t,1}^2 + \hat{\sigma}_{t,2}^2} \right)$$

$$\le \sum_{t=1}^{T_0} \mathbb{P}\left( \tau(1) = t, \tau(2) > T_0, \hat{\mu}_{t,2} - \mu_2 - \hat{\mu}_{t,1} + \mu_1 > \Delta + \kappa \cdot \sqrt{2\log T - 2\log\log T} \cdot \sqrt{\hat{\sigma}_{t,1}^2 + \hat{\sigma}_{t,2}^2} \right)$$

$$\le \sum_{t=1}^{T_0} \mathbb{P}\left( \tau(1) = t, \tau(2) > T_0, \hat{\mu}_{t,2} - \mu_2 > \frac{\Delta}{2} + \frac{\kappa\sqrt{2\log T - 2\log\log T}}{2} \cdot \sqrt{\hat{\sigma}_{t,1}^2 + \hat{\sigma}_{t,2}^2} \right)$$

$$\qquad + \mathbb{P}\left( \tau(1) = t, \tau(2) > T_0, -\hat{\mu}_{t,1} + \mu_1 > \frac{\Delta}{2} + \frac{\kappa\sqrt{\log T - 2\log\log T}}{2} \cdot \sqrt{\hat{\sigma}_{t,1}^2 + \hat{\sigma}_{t,2}^2} \right)$$

$$\le \sum_{t=1}^{T_0} \mathbb{P}\left( \tau(1) = t, \tau(2) > T_0, M_{t,t}(2) > \frac{\Delta}{2}\sum_{s=1}^{t}\sqrt{\pi_{s,2}} + \frac{\kappa\hat{\sigma}_{t,2} \cdot \sqrt{2\log T - 2\log\log T}}{2} \cdot \sum_{s=1}^{t}\sqrt{\pi_{s,2}} \right)$$

$$\qquad + \mathbb{P}\left( \tau(1) = t, \tau(2) > T_0, -M_{t,t}(1) > \frac{\Delta}{2}\sum_{s=1}^{t}\sqrt{\pi_{s,1}} + \frac{\kappa\hat{\sigma}_{t,1} \cdot \sqrt{2\log T - 2\log\log T}}{2} \cdot \sum_{s=1}^{t}\sqrt{\pi_{s,1}} \right)$$

$$\tag{3}$$

For any $a \in \{1, 2\}$ and $t \in [T_0] := \{1, 2, \ldots, T_0\}$, on the event $\{\tau(1) = t, \tau(2) > T_0\}$, $S_{t,a} = M_{t,a}$, and the estimated variance can be bounded as,

$$\hat{\sigma}_{t,a}^2 \cdot \left( \sum_{s=1}^{t} \sqrt{\pi_{s,a}} \right)^2 = \sum_{s=1}^{t} \pi_{s,a} \cdot \left[ (\hat{\Gamma}_{s,a} - \hat{\mu}_{s,a})^2 + 1 \right] \geq \gamma t.$$

As a result,

$$(3) \leq \sum_{t=1}^{T_0} \mathbb{P}\left( \tau(1) = t, \tau(2) > T_0, S_{t,1} > \frac{\Delta\sqrt{\gamma}t}{2} + \frac{\kappa\sqrt{\gamma t} \cdot \sqrt{2\log T - 2\log\log T}}{2} \right)$$

$$+ \mathbb{P}\left( \tau(1) = t, \tau(2) > T_0, S_{t,2} > \frac{\Delta\sqrt{\gamma}t}{2} + \frac{\kappa\sqrt{\gamma t} \cdot \sqrt{2\log T - 2\log\log T}}{2} \right)$$

$$\leq 2 \sum_{t=1}^{T_0} \exp\left( -\frac{\kappa^2\gamma \cdot (2\log T - 2\log\log T)}{32M^2(2 + 2/\sqrt{\gamma} + 1/\gamma) + 8\sigma^2} \right)$$

$$\overset{(a)}{\leq} \frac{2T_0 \left( \log T \right)^3}{T^3} \leq \frac{2 \cdot (\log T)^3}{T^2},$$

where step (a) is due to the choice $\kappa = \sqrt{(48M^2(2 + 2/\sqrt{\gamma} + 1/\gamma) + 12M^2)/\gamma}$. We now proceed to bound $\mathbb{P}(\tau(1) > T_0, \tau(2) > T_0)$. Similarly, here we have

$$\tau(1) > T_0, \tau(2) > T_0 \implies \mathbb{P}(\tilde{r}_{T_0+1,2} > \tilde{r}_{T_0+1,1} \mid \mathcal{H}_{T_0}) > \frac{1}{T}$$

$$\implies \mathbb{P}\left( \frac{\tilde{r}_{T_0+1,2} - \tilde{r}_{T_0+1,1} - \hat{\mu}_{T_0,2} + \hat{\mu}_{T_0,1}}{\kappa\sqrt{\hat{\sigma}_{T_0,2}^2 + \hat{\sigma}_{T_0,1}^2}} > \frac{\hat{\mu}_{T_0,1} - \hat{\mu}_{T_0,2}}{\kappa\sqrt{\hat{\sigma}_{T_0,2}^2 + \hat{\sigma}_{T_0,1}^2}} \,\bigg|\, \mathcal{H}_{T_0} \right) > \frac{1}{T}$$

$$\implies 1 - \Phi\left( \frac{\hat{\mu}_{T_0,1} - \hat{\mu}_{T_0,2}}{\kappa\sqrt{\hat{\sigma}_{T_0,1}^2 + \hat{\sigma}_{t,2}^2}} \right) > \frac{1}{T}$$

$$\implies \frac{\hat{\mu}_{T_0,1} - \hat{\mu}_{T_0,2}}{\kappa\sqrt{\hat{\sigma}_{T_0,1}^2 + \hat{\sigma}_{T_0,2}^2}} \leq \sqrt{2\log T - \log\log T}.$$

Consequently,

$$\mathbb{P}\left( \tau(1) > T_0, \tau(2) > T_0 \right)$$

$$= \mathbb{P}\left( \tau(1) > T_0, \tau(2) > T_0, \hat{\mu}_{T_0,1} - \hat{\mu}_{T_0,2} \leq \kappa\sqrt{2\log T - \log\log T}\sqrt{\hat{\sigma}_{T_0,1}^2 + \hat{\sigma}_{T_0,2}^2} \right)$$

$$\leq \mathbb{P}\left( \tau(1) > T_0, \tau(2) > T_0, \hat{\mu}_{T_0,1} - \mu_1 \leq -\frac{\Delta}{2} + \frac{\kappa\sqrt{2\log T - \log\log T}}{2}\sqrt{\hat{\sigma}_{T_0,1}^2 + \hat{\sigma}_{T_0,2}^2} \right)$$

$$+ \mathbb{P}\left( \tau(1) > T_0, \tau(2) > T_0, \mu_2 - \hat{\mu}_{T_0,2} \leq -\frac{\Delta}{2} + \frac{\kappa\sqrt{2\log T - \log\log T}}{2}\sqrt{\hat{\sigma}_{T_0,1}^2 + \hat{\sigma}_{T_0,2}^2} \right)$$

$$\leq \mathbb{P}\left( S_{T_0,1} \leq -\frac{\Delta}{2}\sum_{s=1}^{T_0}\sqrt{\pi_{s,1}} + \frac{\kappa\sqrt{2\log T - \log\log T}}{2}\sqrt{\hat{\sigma}_{T_0,1}^2 + \hat{\sigma}_{T_0,2}^2}\sum_{s=1}^{T_0}\sqrt{\pi_{s,1}} \right)$$

$$+ \mathbb{P}\left( -S_{T_0,2} \leq -\frac{\Delta}{2}\sum_{s=1}^{T_0}\sqrt{\pi_{s,2}} + \frac{\kappa\sqrt{2\log T - \log\log T}}{2}\sqrt{\hat{\sigma}_{T_0,1}^2 + \hat{\sigma}_{T_0,2}^2}\sum_{s=1}^{T_0}\sqrt{\pi_{s,2}} \right)$$

$$\leq \mathbb{P}\left( S_{T_0,1} \leq -\frac{\Delta}{2}\sqrt{\gamma}T_0 + \frac{\kappa\sqrt{2\log T - \log\log T}}{2}\sqrt{(8M^2/\gamma + 2) \cdot T_0} \right)$$

$$+ \mathbb{P}\left( -S_{T_0,2} \leq -\frac{\Delta}{2}\sqrt{\gamma}T_0 + \frac{\kappa\sqrt{2\log T - \log\log T}}{2}\sqrt{(8M^2/\gamma + 2) \cdot T_0} \right) \quad (4)$$

With the choice of $T_0$,

$$(4) \leq 2 \cdot \mathbb{P}\left( |S_{T_0,1}| \geq 4M \cdot \sqrt{3 + 2/\sqrt{\gamma} + 1/\gamma} \cdot \sqrt{T_0 \log T} \right) \leq 2T^{-2}.$$

Combining everything, we conclude that $\mathbb{P}(E) \geq 1 - 2T^{-2} - T^{-2} \cdot (\log T)^3$.

## A.2 Proof of Theorem 2

Without loss of generality, we let arm 1 be the optimal arm, and define the suboptimality gap for each $a \in \{2, \ldots, K\}$ as:

$$\Delta_a := \mu_1 - \mu_a.$$

Similar to the two-arm case, we define for each $a \in \{2, \ldots, K\}$ the stopping time

$$\tau(a) = \min\{t : p_{t+1,a} < 1/T\},$$

where $p_{t+1,a} = \min_{a' \in \mathcal{A}_t} \mathbb{P}(\tilde{r}_{t+1,a} > \tilde{r}_{t+1,a'} \mid \mathcal{H}_t)$. Define $\kappa_1 = \frac{16MK}{\gamma}$, $\kappa_2 = \frac{16\sqrt{K}M}{\sqrt{\gamma}}$ and $T_0 = 8 \cdot (1 + \sqrt{K/\gamma})^2 \cdot \log T$. In the beginning phase when $t \leq T_0$ we shall use the parameter $\kappa_1$ as $\kappa$, and in the later phase when $t > T_0$, we use $\kappa_2$. Similar to the two-arm case, we define for each $a \in \{2, \ldots, K\}$:

$$T_a := \frac{64M^2 K \log T}{\Delta_a^2 \gamma} \cdot \left(8c_{\max}\sqrt{K/\gamma} + \sqrt{3 + 2\sqrt{K/\gamma} + K/\gamma}\right)^2.$$

Since we consider the regime $\Delta_a \leq M\sqrt{K}$, we have that $T_a \geq T_0$ for all $a \neq 1$.

Moving on, let $E_a := \{\tau(a) \leq T_a, \tau(1) > T_a\}$ for each $a \neq 1$ denote the good event. We now proceed to bound the probability of the good event not happening. To start, note that

$$\mathbb{P}(E_a^c) = \mathbb{P}(\tau(1) \leq T_a) + \mathbb{P}(\tau(a) > T_a, \tau(1) > T_a).$$

The first term can be decomposed as:

$$\mathbb{P}(\tau(1) \leq T_a) = \sum_{t=1}^{T_a} \mathbb{P}(\tau(1) = t) = \sum_{t=1}^{T_a} \mathbb{P}(\tau(1) = t, p_{t+1,1} < 1/T). \tag{5}$$

Above, $p_{t+1,1} < 1/T$ means that there exists $a' \neq 1$ such that $\tau(a') > t$ and $\mathbb{P}(\tilde{r}_{t+1,1} > \tilde{t}_{t,a'} \mid \mathcal{H}_t) < 1/T$. That is,

$$(5) \leq \sum_{t=1}^{T_a} \sum_{a' \neq 1} \mathbb{P}\left(\tau(1) = t, \tau(a') \geq t, \mathbb{P}(\tilde{r}_{t+1,1} > \tilde{r}_{t+1,a'} \mid \mathcal{H}_t) < 1/T\right)$$

$$\leq \sum_{t=1}^{T_0} \sum_{a' \neq 1} \mathbb{P}\left(\tau(1) = t, \tau(a') > t, \hat{\mu}_{t,a'} - \hat{\mu}_{t,1} > \kappa_1 \cdot \sqrt{2\log T - 2\log\log T} \cdot \sqrt{\hat{\sigma}_{t,1}^2 + \hat{\sigma}_{t,a'}^2}\right)$$

$$+ \sum_{t=T_0+1}^{T_a} \sum_{a' \neq 1} \mathbb{P}\left(\tau(1) = t, \tau(a') > t, \hat{\mu}_{t,a'} - \hat{\mu}_{t,1} > \kappa_2 \cdot \sqrt{2\log T - 2\log\log T} \cdot \sqrt{\hat{\sigma}_{t,1}^2 + \hat{\sigma}_{t,a'}^2}\right)$$

$$\leq \sum_{t=1}^{T_0} \sum_{a' \neq 1} \mathbb{P}\left(\tau(a') > t, \hat{\mu}_{t,a'} - \mu_{a'} > \frac{\Delta_{a'}}{2} + \frac{\kappa_1}{2} \cdot \sqrt{2\log T - 2\log\log T} \cdot \sqrt{\hat{\sigma}_{t,1}^2 + \hat{\sigma}_{t,a'}^2}\right)$$

$$+ \mathbb{P}\left(\tau(a') > t, \mu_1 - \hat{\mu}_{t,1} > \frac{\Delta_{a'}}{2} + \frac{\kappa_1}{2} \cdot \sqrt{2\log T - 2\log\log T} \cdot \sqrt{\hat{\sigma}_{t,1}^2 + \hat{\sigma}_{t,a}^2}\right)$$

$$+ \sum_{t=T_0+1}^{T_a} \sum_{a' \neq 1} \mathbb{P}\left(\tau(a') > t, \hat{\mu}_{t,a'} - \mu_{a'} > \frac{\Delta_{a'}}{2} + \frac{\kappa_2}{2} \cdot \sqrt{2\log T - 2\log\log T} \cdot \sqrt{\hat{\sigma}_{t,1}^2 + \hat{\sigma}_{t,a'}^2}\right)$$

$$+ \mathbb{P}\left(\tau(a') > t, \mu_1 - \hat{\mu}_{t,1} > \frac{\Delta_1}{2} + \frac{\kappa_2}{2} \cdot \sqrt{2\log T - 2\log\log T} \cdot \sqrt{\hat{\sigma}_{t,1}^2 + \hat{\sigma}_{t,a}^2}\right)$$

$$\leq \sum_{t=1}^{T_0} \sum_{a' \neq a} \mathbb{P}\left(S_{t,a'} \geq \frac{\Delta_{a'}}{2} \cdot \sqrt{\frac{\gamma}{K}} \cdot t + \frac{\kappa_1\sqrt{\gamma t}}{2\sqrt{K}} \cdot \sqrt{2\log T - 2\log\log T}\right)$$

$$+ \mathbb{P}\left(-S_{t,1} \geq \frac{\Delta_1}{2} \cdot \sqrt{\frac{\gamma}{K}} \cdot t + \frac{\kappa_1\sqrt{\gamma t}}{2\sqrt{K}} \cdot \sqrt{2\log T - 2\log\log T}\right) \tag{6}$$

$$+ \sum_{t=T_0+1}^{T_a} \sum_{a' \neq a} \mathbb{P}\left(S_{t,a'} \geq \frac{\Delta_{a'}}{2} \cdot \sqrt{\frac{\gamma}{K}} \cdot t + \frac{\kappa_2\sqrt{\gamma t}}{2\sqrt{K}} \cdot \sqrt{2\log T - 2\log\log T}\right)$$

$$+ \mathbb{P}\Big(-S_{t,1} \geq \frac{\Delta_{a'}}{2} \cdot \sqrt{\frac{\gamma}{K}} \cdot t + \frac{\kappa_2 \sqrt{\gamma t}}{2\sqrt{K}} \cdot \sqrt{2\log T - 2\log\log T}\Big). \tag{7}$$

Recall that we define the martingale difference as

$$D_{t,a} = S_{t,a} - S_{t-1,a} = \mathbf{1}\{\tau(a) \geq t\} \cdot \sqrt{\pi_{t,a}} \cdot (\hat{\Gamma}_{t,a} - \mu_a).$$

We then have the following bounds:

$$\sum_{s=1}^{t} D_{s,a}^2 \leq 4M^2 \cdot \Big(1 + \sqrt{\frac{K}{\gamma}}\Big)^2 \cdot t;$$

$$\sum_{s=1}^{t} \mathbb{E}\big[D_{s,a}^2 \mid \mathcal{H}_{s-1}\big] \leq (4M^2 + \sigma^2) \cdot t.$$

Before proceeding, we state in Lemma 3 the Freedman's inequality that we shall use in the subsequent analysis.

**Lemma 3 (Freedman's inequality)** *Let $\{M_n\}_{n\geq 1}$ be a locally square integrable real martingale w.r.t. $\{\mathcal{F}_n\}_{n\geq 1}$ such that, for each $1 \leq k \leq n$, $|M_k - M_{k-1}| \leq c$ a.s. for some constant c. Then, for all $x, y > 0$,*

$$\mathbb{P}\Big(M_n \geq x, \sum_{i=1}^{n} \mathbb{E}\big[(M_i - M_{i-1})^2 \mid \mathcal{F}_{i-1}\big] \leq y\Big) \leq \exp\Big(-\frac{x^2}{2\cdot(y+cx)}\Big).$$

Applying Lemma 1 to (6) for the terms with $t \leq T_0$, we have that

$$\mathbb{P}\Big(-S_{t,1} \geq \frac{\Delta_1}{2} \cdot \sqrt{\frac{\gamma}{K}} \cdot t + \frac{\kappa_1 \sqrt{\gamma t}}{2\sqrt{K}} \cdot \sqrt{2\log T - 2\log\log T}\Big) \leq \exp\Big(-\frac{\gamma \kappa_1^2 \cdot (\log T - \log\log T)}{16M^2 K(3 + 2\sqrt{K}/\sqrt{\gamma} + K/\gamma)}\Big).$$

Alternatively, applying Lemma 3 to (6) for the terms with $t > T_0$ yields

$$\mathbb{P}\Big(S_{t,1} \geq \frac{\Delta_{a'}}{2} \cdot \sqrt{\frac{\gamma}{K}} \cdot t + \frac{\kappa_2 \sqrt{\gamma t}}{2\sqrt{K}} \cdot \sqrt{2\log T - 2\log\log T}\Big)$$

$$\leq \exp\Big(-\frac{\kappa_2^2 \gamma t(\log T - \log\log T)}{4K\big(8M^2 t + (1 + \sqrt{K/\gamma})M\kappa_2\sqrt{\gamma t}\sqrt{(2\log T - 2\log\log T)/K}\big)}\Big).$$

Combining the above two inequalities, we have

$$(6) \leq \sum_{s=1}^{T_0} \exp\Big(-\frac{\gamma \kappa_1^2 \cdot (\log T - \log\log T)}{16M^2 K(3 + 2\sqrt{K}/\sqrt{\gamma} + K/\gamma)}\Big) + \sum_{s=T_0+1}^{T_a} \exp\Big(-\frac{\gamma \kappa_2^2 (\log T - \log\log T)}{64KM^2}\Big)$$

$$\leq 2 \cdot T_a \cdot \Big(\frac{\log T}{T}\Big)^4 \leq 2 \cdot \frac{(\log T)^4}{T^3}.$$

Switching to the second term, for $a \in \{2, \ldots, K\}$, we have that

$$\mathbb{P}\Big(\tau(a) > T_a, \tau(1) > T_a\Big)$$

$$\leq \mathbb{P}\Big(\tau(a) > T_a, \tau(1) > T_a, \mathbb{P}(\tilde{r}_{T_a+1,a} > \tilde{r}_{T_a+1,1} \mid \mathcal{H}_{T_a}) > 1/T\Big)$$

$$\leq \mathbb{P}\Big(\tau(a) > T_a, \tau(1) > T_a, \hat{\mu}_{T_a+1,a} - \hat{\mu}_{T_a+1,1} > -\kappa_2\sqrt{\hat{\sigma}_{T_a,1}^2 + \hat{\sigma}_{T_a,a}^2}\sqrt{2\log T - \log\log T}\Big)$$

$$\leq \mathbb{P}\Big(\tau(a) > T_a, \tau(1) > T_a, \hat{\mu}_{T_a+1,a} - \mu_a > \frac{\Delta_a}{2} - \frac{\kappa_2}{2}\sqrt{\hat{\sigma}_{T_a,1}^2 + \hat{\sigma}_{T_a,a}^2}\sqrt{2\log T - \log\log T}\Big)$$

$$\quad + \mathbb{P}\Big(\tau(a) > T_a, \tau(1) > T_a, \mu_1 - \hat{\mu}_{t+1,1} > \frac{\Delta_a}{2} - \frac{\kappa_2}{2}\sqrt{\hat{\sigma}_{T_a,1}^2 + \hat{\sigma}_{T_a,a}^2}\sqrt{2\log T - \log\log T}\Big)$$

$$\leq \mathbb{P}\Big(S_{T_a,a} > \frac{\Delta_a T_a}{2}\sqrt{\frac{\gamma}{K}} - 2c_{\max}\kappa_2\sqrt{T_a}\sqrt{2\log T - \log\log T}\Big)$$

$$+ \mathbb{P}\Big( -S_{T_a,1} > \frac{\Delta_a T_a}{2}\sqrt{\frac{\gamma}{K}} - 2c_{\max}\kappa_2\sqrt{T_a}\sqrt{2\log T - \log\log T}\Big) \tag{8}$$

Applying Lemma 1 and with the choice of $T_a$, we arrive at (8) $\leq T^{-2}$. Finally, denoting the number of pulls of arm $a$ by $N(a)$, we decompose the regret as

$$\mathbb{E}\big[R(T,\pi)\big] = \sum_{a=2}^{K}\Delta_a\mathbb{E}\big[N(a)\big]$$

$$= \sum_{a=1}^{K}\Delta_a\Big(\mathbb{E}\big[N(a)E_a\big] + \mathbb{E}\big[N(a)E_a^c\big]\Big)$$

$$\leq \sum_{a=2}^{K}\Delta_a\sqrt{T_a}\cdot\mathbb{E}\big[\sqrt{N(a)}\big] + o(1)$$

$$\leq O(K\sqrt{\log T})\cdot\sum_{a=2}^{K}\mathbb{E}\big[\sqrt{N(a)}\big] + o(1)$$

$$\leq O(K^{3/2}\sqrt{T\log T}),$$

where the last inequality is due to $\frac{1}{K-1}\sum_{a=2}^{K}\sqrt{N(a)} \leq \sqrt{\frac{1}{K-1}\sum_{a=2}^{K}N(a)} \leq \sqrt{T/(K-1)}$.

## B  Additional Empirical Investigation

### B.1  Drawbacks of sample average, IPW and DR in a toy adaptive setting

In this subsection, we visualize the bias of the sample average estimator and the lack of asymptotic normality of the IPW and DR estimators when used in adaptively collected data. To do so, we reproduce the toy adaptive domain presented in [Hadad et al., 2019]. This toy adaptive domain is a two-stage, two-arm trial where $r_t(a) \sim \mathcal{N}(0,1)$ for both arms $a = 0, 1$. For the first $T/2$ time periods, arm selection is randomized with probability 50%. After $T/2$ time periods, the arm with the largest sample average is identified, and for the next $T/2$ time periods it is allocated with probability 90% whereas the other arm is allocated with probability 10%. We set $T = 20,000$ and repeat this toy adaptive domain for $100,000$ simulations.

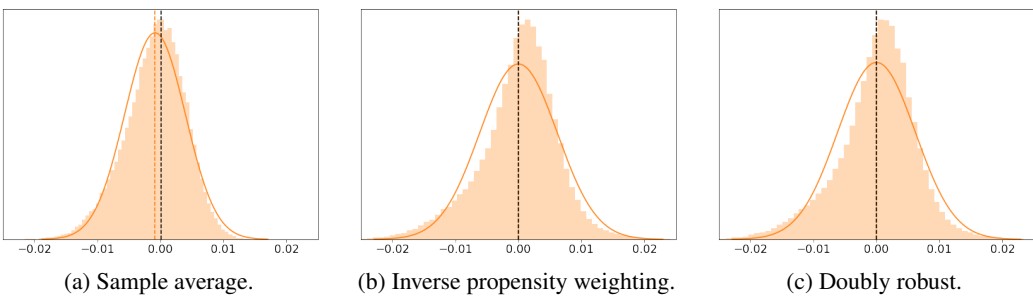

|(a) Sample average.|(b) Inverse propensity weighting.|(c) Doubly robust.|

Figure 3: Distribution of the estimators $\hat{Q}^{\text{SA}}$, $\hat{Q}^{\text{IPW}}$ and $\hat{Q}^{\text{DR}}$ as described in section 3.1 applied to a toy adaptive domain. The histogram depicts the distribution of the estimators over $100,000$ simulations. The red dashed line corresponds to the estimate averaged across simulations, whereas the black dashed line corresponds to the ground truth. The red solid line corresponds to the normal distribution matching the first two moments of the estimator histogram.

In figure 3, we see the histogram of the estimates of arm's $a = 1$ true mean from these $100,000$ simulations at the end of the adaptive data collection based on the sample average, IPW and DR estimators. As described in section 3.1 and seen in figure 3, sample average has negative bias in adaptively collected data, because arms for which we observe random downward fluctuations will be sampled less and the negative bias will not be corrected, whereas arms for which we observe

random upward fluctuations will be sampled more and the positive bias will be corrected, contributing to the overall negative bias of the estimator. Also, the distribution of the sample average is not asymptotically normal. IPW and DR correct the bias due to the adaptive data collection but in doing so they exacerbate the lack of asymptotic normality of the estimator, as seen in figure 3.

## B.2 Sensitivity of algorithms to the propensity-controlling parameters

We use the low SNR setting of the semisynthetic experiment described in Section 4.2 of the main paper to explore the sensitivity of the "causal" TS algorithms which use unbiased estimators $\hat{Q}_{t,a}^{\text{IPW}}$ (TS-IPW), $\hat{Q}_{t,a}^{\text{DR}}$ (TS-DR) and $\hat{Q}_{t,a}^{\text{ADR}}$ (DATS) to the choice of parameter $\gamma$, which controls the amount of uniform exploration over non-eliminated arms $\mathcal{A}_t$ at time $t$ and consequently how small the propensity scores get. In the case of the heuristic, DATS-clipping, the propensity controlling parameter is $\delta$ and is merely a clipping threshold of the propensity score in the $\hat{Q}_{t,a}^{\text{ADR}}$ estimator (rendering it no-longer unbiased). We try uniform-explore values $\gamma = 0.01, 0.05, 0.1$ for TS-IPW, TS-DR and DATS and clipping values $\delta = 0.001, 0.01, 0.02$ for the heuristic DATS-clipping. In Figure 4, we see that both DATS and its heuristic DATS-clipping are robust to the choice of propensity-controlling parameters, whereas TS-IPW and TS-DR which use the non-adaptively-weighted and high-variance estimators $\hat{Q}_{t,a}^{\text{IPW}}$ and $\hat{Q}_{t,a}^{\text{DR}}$ instead of $\hat{Q}_{t,a}^{\text{ADR}}$ are very sensitive to it. We observe that by tuning $\gamma$, TS-IPW and TS-DR can improve their performance and even become competitive to TS and UCB, whereas with the default parameter TS-IPW and TS-DR were under-performed by TS and best-tuned UCB due to their poorly-controlled variance. On the other hand, thanks to the variance stabilization properties of the adaptive weights, DATS and DATS-clipping remain the best performing variants even among the best-tuned TS-IPW and TS-DR.

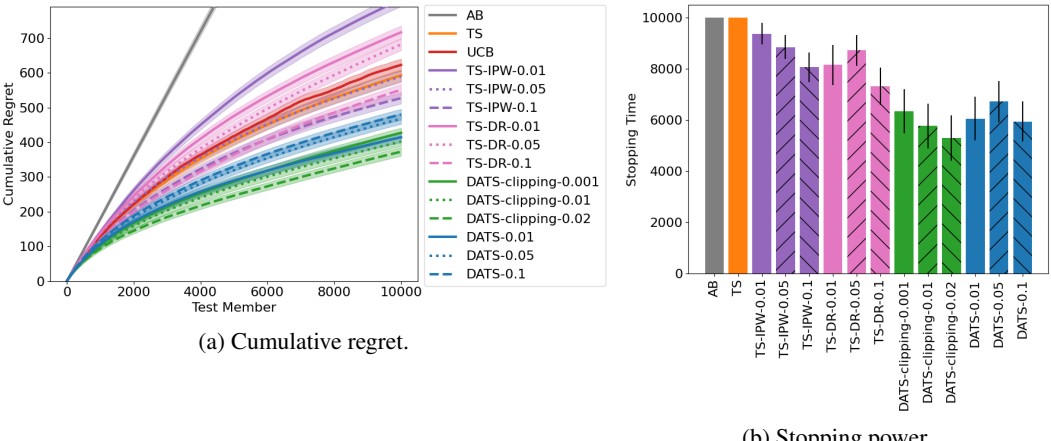

(a) Cumulative regret.

(b) Stopping power.

Figure 4: Sensitivity of TS-IPW, TS-DR, DATS and DATS-clipping to propensity-controlling parameters in terms of cumulative regret and stopping power in the low SNR setting of Section's 4.2 semi-synthetic experiment