# OpenReview forum: "Online Multi-Armed Bandits with Adaptive Inference"
_NeurIPS.cc/2021/Conference — NeurIPS 2021 Poster_

### Official Review · Reviewer_zzib · 2021-07-03

**Rating:** 7
**Confidence:** 3

**Summary:**

The paper proposes a Thompson Sampling-based online algorithm for the stochastic K-armed bandit problem, which exhibits improved numerical performance in terms of the expected cumulative regret and its variance, and smaller stopping times for best arm identification. The improvement is over the popular UCB and Thompson Sampling baselines common in literature. Although, the improved empirical performance of the proposed algorithm does not translate to provably better regret guarantees vis-a-vis the aforementioned benchmarks (a sub-optimal K-dependent scaling factor is present in the upper bound). The improved numerical performance is primarily attributed to a new estimator for the true arm-means (as opposed to the standard sample-average approximation used by vanilla UCB/TS), which is provably unbiased, and at the same time, offers a better variance control compared to other unbiased estimators popular in ex post (adaptive) causal inference literature. The punchline, in my understanding, is that it is possible to improve and (better) control the numerical performance of bandit algorithms by employing sophisticated estimators from adaptive inference literature, without compromising on rate-optimality in the regret upper bound (order of dependence on the sampling horizon, modulo multiplicative constants).

**Limitations And Societal Impact:**

Some broad remarks and minor typographical errors are noted below:

1. Line 34: It should be "..... uncertainty," (the order of " and , should be flipped). There are other occurrences too.

2. Line 37,38: This is a bit ambiguous; gap-dependent bounds are not the same as "worst-case" bounds in standard bandit parlance.

3. Footnote 2: Also state the $O\left( \sqrt{KT\log K} \right)$ bound for the Gaussian prior-based version of TS from the same reference, for a comprehensive picture (the stated bound corresponds to the Beta prior-based version of TS).

4. Footnote 3: This is not true for UCB. Previously cited paper dispels this myth (Refer to Theorem 1 and 4 therein).

5. Line 114: Policy-dependence of regret should be made explicit on the RHS.

6. Line 144-145: Same as above, i.e., claim re asymptotic-(non)normality needs to be revisited and corrected for UCB.

7. In Algorithm 1 "arm elimination step," the first update needs to happen for each $a\in\mathcal{A}_t$.

8. In Algorithm 1 "propensity score computation" step, it should be $\tilde{r}_{t+1,a}$ after "argmax."

9. Line 200: "s" should be "t," and it should be "a" in place of $\tilde{r}_{t,a}$.



**Main Review:**

The paper is well-written overall with intuitive explanations, and an accessible proof of the main theoretical result. The technical development occurs at the intersection of adaptive inference with the traditional regret viewpoint in the learning literature, and certainly seems to be of practical significance (numerics are promising). Also, extant literature in this space is relatively recent and sparse (to the best of my knowledge), making this work even more appealing. The literature review seems reasonably complete, although the following paper is missing: https://arxiv.org/pdf/2106.02126.pdf. This is a recent paper that answers some long standing open questions regarding the distribution of arm-pulls under UCB, TS and can significantly shape/alter the discussion in the current paper regarding asymptotic-normality (or lack thereof) under vanilla UCB/TS. Cited paper, I think, might be able to provide a more comprehensive and theoretically grounded justification for why it may be advisable to move towards a greater level of sophistication in estimator design in bandit settings.

**Time Spent Reviewing:**

6

---

> ### Author Response · Authors · 2021-08-10
> **Authors' Response to Reviewer zzib**
>
> Thank you for taking the time to review the paper. We are glad you appreciated both the novelty and value of the results and the clarity of exposition and comparison. We worked hard to communicate our results clearly, correctly, and concisely in order to inspire further research in the intersection of these two fields.
>
> Also, thank you for highlighting this citation https://arxiv.org/pdf/2106.02126.pdf. It is indeed an important advancement which is very relevant to discuss in our paper, and although of course we could not have discussed it at the time of submission as it became public shortly after, we will certainly add a discussion on it in our revision. Indeed, understanding of the distribution of arm-pulls, as characterized by that paper, helps us motivate the necessity of constructing more sophisticated estimators for better inference in the online bandit domain, which is a connection often missed in practice as the online bandit literature and the offline inference literature seem to progress in parallel with few bridges between them. The cited paper is one of these bridges, so it is a very relevant citation to include in our paper which aims to contribute an additional bridge.
>
> Finally, thank you for identifying typos and making the corrections below. This indicates a close and concentrated reading for which we are very grateful. We respond below on how we will fix these in the order they appear.
> 1. We have taken a pass and fixed these typos accordingly.
> 2. We have modified the sentence to “The finite-time regret bounds for these algorithms–$\Theta(\log{T})$ gap-dependent bounds and $\Theta(\sqrt{T})$ gap-independent bounds–are both optimal, with the latter in the worst-case sense.”
> 3. We have added the bound for the Gaussian prior-based version in the footnote.
> 4. Thank you for highlighting this and sending us the citation of the paper. We will correct the statement based on that recently-published work.
> 5. We have modified it to indicate that $a_t \sim \pi_t$ in the RHS.
> 6. Thank you, we will address the same way as answer 4.
> 7. That is correct and we have made it explicitly by adding $\forall a \in \mathcal{A}_t$ before “and” on that line.
> 8. Fixed.
> 9. Fixed.

---

### Official Review · Reviewer_wmKw · 2021-07-12

**Rating:** 4
**Confidence:** 4

**Summary:**

The authors propose a Thompson Sampling (TS) based cumulative regret-minimsation algorithm for stochastic MAB. Their approach pivots on the unbiased estimator of mean using inverse propensity score, and then it applies weighted averaging scheme [Luedtke and Van Der Laan, 2016] to reduce the variance of the estimate. Further, they establish theoretical worst-case upper bound on the expected cumulative regret, and present multiple simulations to show empirical efficiency of their proposed algorithm.

**Limitations And Societal Impact:**

Mention of limitation of the proposed approach is not apparent.

**Main Review:**

below is the list of strengths ad weaknesses of the paper:

Strengths:
1) The proposed algorithm is empirically efficient.
2) Use of propensity score along with weighted averaging for a better estimation of the mean is a god idea.


Weaknesses:
1)  It seems that to hold the proposed approach, the reward distribution of the individual arms has to be Gaussian; however, there is no explicit mention of the same. There is no discussion on the family of sub-gaussian reward distribution.
2) The scope of this proposed idea for sample averaging by propensity score, does not seem to hold for paradigm like UCB, wherein, given the history, the next arm to be pulled is obtained deterministically.
3) The problem-independent upper bound on cumulative regret presented in Theorem 2 is far from the optimal rate O(\sqrt(KT)).

Overall the contribution appears to be marginal.

Minor Comment:
There is only a single subsection 1.1 under the section 1.

**Time Spent Reviewing:**

5

---

> ### Author Response · Authors · 2021-08-10
> **Authors' Response to Reviewer wmKw**
>
> Thank you for taking the time to review our paper and we appreciate your positive comments regarding the strengths of our paper. Please see our responses below to the points you raise in the order they appear.
>
> 1. Thanks for raising this question and we see the confusion we might have caused in our exposition. First, just to be clear, we do not assume in establishing the theorem that the reward is Guassian. The only assumption needed is that the reward is bounded. Now, in hindsight, we do see why you raised this question: in line 200 of the paper and second to last line of Algorithm 1, we sampled the estimated true mean of an arm from a Guassian distribution. We clarify here that using this Guassian posterior distribution is a procedure in our algorithm, rather than an assumption we make. In other words, the actual reward distribution need not be Guassian, and our regret bound still holds under Algorithm 1. We will emphasize this in our paper.
>
> 2. Thanks for raising this point. Indeed, our approach requires the algorithm to have a positive propensity score at each time step (i.e. each action under consideration has a non-zero probability of being selected). Hence UCB-type algorithms do not immediately fit into this framework, and that is also the case for related de-biasing approaches in the offline adaptive inference literature that require non-trivial propensities as well (e.g., van der Laan, 2008; Luedtke & van der Laan 2016; Hadad et al. 2019, Zhang et al. 2021 among others). So, our method is better suited to be used with stochastic exploration algorithms, such as TS which has non-trivial (0 or 1) propensities that we know in each time period due to the probability-matching nature of TS. That said, UCB could potentially benefit from the proposed approach given that it also uses sample averages in the confidence bound: from an algorithmic point of view, one can use historical data on how frequently a particular arm has been pulled as a “proxy” for its propensity score, to be plugged into the estimator. Of course, this would be an entirely different inquiry beyond this paper on a different class of algorithms that are non-stochastic.
>
> 3. Following a suggestion by Reviewer oqBT, in our revised manuscript, we improved the dependence in $K$ by a factor of $\sqrt{K}$.
> Indeed, (even after the improvement) our regret bound is not optimal in the number of arms $K$, but is minimax optimal (up to log factors) in the horizon $T$. We point out that we do not attempt to be tight in $K$ in our bound, as in the applications we have in mind (e.g. clinical trials, web-service testing), including the one motivating our semi-synthetic experiment, the number of arms is typically a (small) constant. That said, we believe that the suboptimal dependence on $K$ is an artifact of the proof, rather than the property of the algorithm (Figure 1 in the paper demonstrates empirically that the dependence of the regret on $K$ is of the order $\sqrt{K}$ for 20 synthetic domains with $K$ ranging from 5 to 50 and high/low noise settings). We leave even tighter analysis w.r.t. to $K$ as future work and we hope that this does not deter the reader from the main contributions of our paper.
>
> With regards to the overall strength of our submission, respectfully, we believe that the paper makes important contributions to the community:
> 1. We have highlighted an area for improvement which lies at the core of many online bandit algorithms, which is the need for more sophisticated estimation of an arm’s mean reward to complement their efficient exploration schemes in the online setting. We have made a conceptual advance by connecting two literatures -- that of online learning and that of offline inference -- which have generally been progressing in parallel and contributions in their intersection have been very sparse, to the best of our knowledge.
> 2. With our proposed algorithm, we make the point that one can go beyond worst-case regret performance and that by bringing well-developed ideas from the offline inference literature in online bandit algorithms, we can unlock further performance gains compared to TS and UCB which are wide-spread in practice and theoretically optimal in the worst-case. The empirical study in several synthetic and semi-synthetic domains corroborate the practical advantage of such an algorithm.
> 3. All that while proving that this type of bandit algorithm can achieve this superior practical performance without giving away regret guarantees which are minimax optimal (up to log factors) in the horizon $T$.  Notably, all existing results on adaptive estimators have been asymptotic in nature (typically in the style of CLT bounds), whereas our bound is finite in nature and sheds light in the arena of using these for online decision making, rather than offline inference, on which all prior work on adaptive estimators has focused–to our knowledge.
> 4. As such, we have hoped to inspire more research in the intersection of these two fields that will generate more algorithms that perform better in practice while maintaining their theoretical soundness.
>
> We worked hard to convey the above messages in a clear and concise manner and we would be happy to emphasize them even further in the paper, as we have done here.

---

### Official Review · Reviewer_gbUL · 2021-07-16

**Rating:** 8
**Confidence:** 5

**Summary:**

The paper investigates stochastic MAB algorithms, UCB and Thompson sampling, where the usual choice for estimating the means of arms, importance weighting, is replaced with a "doubly adaptive" estimator. This estimator, proposed by Leudtke and Van Der Laan 2016, is self-normalized in that the weight of each sample in the mean estimator is not uniform but rather proportional to the inverse square root propensity weight; it also uses an AIPW-style estimator with these weights.

The authors claim that this new estimator can have better practical performance, since the variance of the mean estimates is greatly reduced, and this claim is verified with empirical simulations. Additionally, the authors provide an upper bound on the worst case performance of O(K^2 sqrt{T log T}), which captures the correct dependence in T.


**Limitations And Societal Impact:**

yes

**Main Review:**

This is a great paper. First, it studies a archetypal NeurIPS problem, the stochastic MAB, and asks an important and very neglected question: what can we say beyond worst-case performance? Second, it brings in powerful and well developed ideas from other literature that are ignored by our community. Third, it is well written, cohesive, and easy to follow. The inclusion of the two-arm upper-bound proof in the main body is appreciated, since it gets the main points of the analysis across without taking up too much space.

There are a few weaknesses, besides the K^2 term in the upper bound. There is no theory supporting the central hypothesis that DATS can have improved sample complexity. Maybe a particular instance where a gap can be shown between the IPW and a DATS-based algorithm is possible?

What is the point of the \Phi is algorithm 1 (it is monotonic)?

post-rebuttal: my opinion of this paper remains high and my score unchanged.

**Time Spent Reviewing:**

3

---

> ### Author Response · Authors · 2021-08-10
> **Authors' Response to Reviewer gbUL**
>
> Thank you for taking the time to review our paper and for your encouraging comments. We were very glad to read that you appreciated the importance of the problem and the value of our contribution and that you found the paper well-written. We appreciate your helpful feedback and let us respond to your questions in order.
>
> - You are raising an excellent point that the investigation of similar ideas in the best-arm identification literature and under the sample complexity measure is a very promising avenue of future work. Even though--as you correctly point out--TS, TS-IPW, TS-DR and DATS are not optimized and analyzed under the sample complexity measure but under the regret measure, we thought it would be interesting to show empirically how TS, TS-IPW, TS-DR and DATS compare under sample complexity additionally to regret. The motivation behind this is that many applications, such as testing in web-services, have as primary goal the test member experience but as a secondary goal how quickly the best arm can be identified even when using a regret-minimization algorithm. As such, we thought it would be informative to show that DATS appears advantageous to other regret-minimization-baselines in terms of that goal as well.
>
>     That said, you are raising an excellent point regarding the application of the same technique to best-arm identification algorithms. We believe that such algorithms, like Top-Two Thompson sampling (TTTS), which is optimized for the sample complexity measure, could also benefit from adaptive inference as well since the usage of the sample averages is at its core as well. E.g., the propensity score of TTTS  is a function of the propensity score of TS sampling (Russo 2016: “Simple Bayesian Algorithms for Best-Arm Identification”), so the benefits of DATS in the multi-armed bandit setting that we discussed in this paper are directly transferable and could motivate the DA-TTTS best-arm identification algorithm. Under similar arguments, correcting the bias of the sample averages used in TTTS with estimators such as IPW and DR will not be enough without the presence of stabilization weights that also have desirable variance properties as well and the normality guarantee. So, one would expect -- as you correctly point-out -- that TTTS-IPW and TTTS-DR would be outperformed by DA-TTTS. So, the application of this idea in the best-arm identification literature and the corresponding investigation under a sample complexity measure is an excellent source of research, which we leave for our future work.
>
> - Thanks for the clarification question. $\Phi$ denotes the CDF of a standard normal distribution and it enters the algorithm for the arm elimination step. We will make this clear in the paper. Its purpose is to compute the probability that a sample from the posterior of arm $a$ is greater than a sample from the posterior of arm $a’$. The difference $D_{a,a',t}$ of the sample of arm $a$ minus the sample of arm $a'$ at time $t$ is distributed as $D_{a,a',t} \sim \mathcal{N}(\mu_{a,t} - \mu_{a',t}, \sigma^2_{a,t} + \sigma^2_{a',t})$. If there exists an arm $a'$ for which the probability $P(D_{a,a',t} > 0)$ becomes smaller than $1/T$, then arm $a$ is eliminated. To compute this probability we use the CDF of the standard normal distribution $P(D_{a, a', t} > 0) = P(-D_{a, a', t} < 0) = \Phi(\frac{0 - (-(\mu_{a, t} - \mu_{a’, t}))}{\sigma^2_{a,t} + \sigma^2_{a',t}}) = \Phi(\frac{\mu_{a,t} - \mu_{a',t}}{\sigma^2_{a,t} + \sigma^2_{a',t}})$, as in Algorithm 1. We will add this explanation in our revision.

---

### Official Review · Reviewer_oqBT · 2021-07-16

**Rating:** 6
**Confidence:** 3

**Summary:**

This paper considers the multi-arm bandit problem. It starts from the observation (already present in the literature) that classical bandit algorithms such as UCB or TS estimate the mean rewards of the arms by simple empirical averages without taking into account the fact that they are adaptively sampled and are therefore biased. The paper thus proposes to make these estimators unbiased by using importance weighting (inverse propensity score weighting) as well as variance reduction tools (adaptive doubly robust estimator and clipping). Applying it to TS, they empirically demonstrate that their algorithm significantly outperforms classical TS and UCB while maintaining a theoretical worst-case regret bound of order $O(\sqrt{T})$.

**Limitations And Societal Impact:**

Yes

**Main Review:**

The empirical contribution of the paper is interesting and having bandit algorithms with improved performance is significant. The simulation results in this paper seem compelling to me. While the application of the inverse propensity score is not new in the bandit literature, variance reduction techniques have been shown to perform very well in counterfactual learning and optimization and I am not aware of any previous work applying it to TS or UCB. It is instructive that this also applies to multi-armed bandits.


On the negative side of the paper:
1. I did not find it so well written.
	- Probably because of the page constraint, the heavy use of inline formulas makes the reading sometimes difficult.
	- In addition, I would have liked more explanations and informal proofs for some of the claims: why is the standard empirical average biased and what is the bias? I agree that the authors explain that this is due to adaptive sampling and that this has been shown in previous work. But this statement is important to the message of the paper and would make the paper more self-contained. Similarly, a simple formula that shows that the new doubly robust adaptive estimator is indeed unbiased and that its variance is less than that of DR or IPW might help the reader who is not familiar with counterfactual learning. At least the latter claims could probably be illustrated in a figure with synthetic data.

2. Also, the regret bound is suboptimal in the number of arms K (of order $K^2\sqrt{T}$ instead of $\sqrt{KT}$). Some parts of the analysis could probably be improved. For example, the upper bound of $N_a$ by $T_a$ in the last equation of p16 seems very suboptimal and gives an extra K factor. Is it not possible to improve this?

Other comments:
- I agree that the method seems more suitable to TS to compute inverse propensity scores. But do you think that it could also be applied to UCB based algorithm to improve their estimates?
- I would also write in the main text that you also provide distribution dependent regret bound in $O(\log T / \Delta)$
- For best-arm identification simulations, it could be interesting to show the frequency of success of all algorithms. It is supposed to be approximately 0.95 in theory, but what is it in practice?
- What motivates the default choice gamma = 0.01 for the algorithm? Doesn't it yield linear regret? Shouldn't it depend on K in practice?
- TS and UCB are not the standard algorithm for best arm identification. It could be interesting to apply the techniques to best-arm identification algorithms.
- How is exactly AB implemented in the simulations?
- I suspect that the regret bound of Thm. 1 should be $\sqrt{T}log T$ and not $\sqrt{T log T}$. Indeed, $T_0$ as defined in line 514 is of order $(\log T / \Delta)^2$ and not $\log(T)/Delta^2$ as stated in line 303.
- Since the bound is of order $\sqrt{T}$, I do not find Cor. 1 very interesting since it is a straightforward application of Azuma-Hoeffding inequality to Thm. 1. I would replace it with a single-line comment in the text stating that the bound also holds with h.p.


Typos:
- l92: a dot is missing
- l78: I would define IPW there
- l307: a T should not be there

**Time Spent Reviewing:**

3

---

> ### Author Response · Authors · 2021-08-10
> **Authors' Response to Reviewer oqBT**
>
> Thank you for your review and your encouraging comments. We are glad to hear that you found our contribution valuable to the community and our empirical results strong. We respond to your questions and feedback below in the order they appear in the review.
>
> Main Comments:
> 1. Thank you for helpfully pointing out places where we can improve clarity. While we worked hard to communicate the motivation and results in a clear, intuitive and concise manner, your feedback is very helpful to further improve it and we respond below on how we will address it.
>     - Thanks for the observation regarding inline formulas. We will limit their use.
>     - Thanks for the comment regarding intuitive explanations. Per your suggestion, we will add the following discussion points and visualization in order to enhance intuition.
>         - First, that for the sample average, there is negative bias in the adaptive setting because arms that are “unlucky” and their sample average is lower than their true mean are sampled less and this negative bias is not corrected, while arms that are “lucky” and their sample average is higher than their true mean are sampled more and this positive bias is corrected, accounting for a negative bias of the estimator overall.
>         - Second, that for IPW and DR using importance sampling weights based on true propensity scores (true because in an online setting we control and know the policy) removes the bias but contributes to the lack of asymptotic normality. This lack of asymptotic normality comes from the fact that in an adaptive setting the importance-sampling ratios converge to 1 or diverge to infinity. As a result, both estimators' variance may either be dominated by their first terms or their last terms. At a more theoretical level, in an adaptive setting this violates the classical condition of martingale central limit theorems that the conditional variance of the terms given previous observations stabilizes asymptotically. On the other hand, the stabilization weights of the ADR estimator equalize the contribution of each datapoint to the variance of the estimator.
>         - Finally, the visualization idea you suggest is a great way to capture the differences of these succinctly. We will implement this in a simple synthetic domain (e.g. 2 arms) and show that by applying the various estimators over multiple simulations we would observe that the sample average has a histogram which is below 0 (due to its negative bias), while IPW and DR have histograms which are centered around 0 (unbiased) but they are wider due to the higher variance and their lack of normality due to the diverging importance-sampling weights is very apparent. And, ADR’s histogram is both centered around 0, tighter and normal. We will include this visualization (effectively an expanded version of Figure 1 in Hadad et al. 2019) on synthetic data in our paper to enhance understanding.
>
> 2. Thank you for raising this point and for the great suggestion. Per your suggestion, in our revised manuscript, we conduct a more careful analysis in the multi-arm case: 1) we introduce the Freedman’s inequality to improve the tightness of the concentration inequality (Freedman’s inequality is analogous to the Berstein’s inequality, but works with adaptive data); 2) additionally, when bounding $N_a$, we use the upper bound $\sqrt{N_a} \cdot \sqrt{T_a}$ instead of $T_a$ (the summation of the $\sqrt{N_a}$ part can be handled by the Jensen’s inequality). The above changes together improve the bound by a factor of $\sqrt{K}$.
>
>     Indeed, (even with the improvement) our regret bound is not optimal in the number of arms $K$, but is minimax optimal (up to log factors) in the horizon $T$. We point out that we do not attempt to be tight in $K$ in our bound, as in the applications we have in mind (e.g. clinical trials, web-service testing), including the one motivating our semi-synthetic experiment, the number of arms is typically a (small) constant. That said, we believe that the suboptimal dependence on $K$ is still an artifact of the proof, rather than the property of the algorithm (Figure 1 in the paper demonstrates empirically that the dependence of the regret on $K$ is of the order $\sqrt{K}$ for 20 synthetic domains with $K$ ranging from 5 to 50 and high/low noise settings). We leave even tighter analysis w.r.t. to $K$ as future work and we hope that this does not deter the reader from the main contributions of our paper.
>
> Other Comments:
> - Thanks for raising this point. Indeed, our approach requires the algorithm to have a positive propensity score at each time step (i.e. each action under consideration has a non-zero probability of being selected). Hence UCB-type algorithms do not immediately fit into this framework, and that is also the case for related de-biasing approaches in the offline adaptive inference literature that require non-trivial propensities as well (e.g., van der Laan, 2008; Luedtke & van der Laan 2016; Hadad et al. 2019, Zhang 2021 among others). So, our method is better suited to be used with stochastic exploration algorithms, such as TS which has non-trivial (0 or 1) propensities that we know in each time period due to the probability-matching nature of TS. That said, UCB could potentially benefit from the proposed approach given that it also uses sample averages in the confidence bound: from an algorithmic point of view, one can use historical data on how frequently a particular arm has been pulled as a “proxy” for its propensity score, to be plugged into the estimator. Of course, this would be an entirely different inquiry on a different class of algorithms that are non-stochastic.
> - Thank you, we will add this note.
> - Yes, it is close to 95% in practice as well, but DATS identifies the optimal arm with fewer samples.
> - Thanks for the clarification question and we will make it clear in the paper that the presence of $\gamma$ does *not* yield linear regret due to the existence of the arm elimination step in our algorithm and the fact that the algorithm applies uniform exploration of magnitude $\gamma$ at time $t+1$ *οnly* over the set of the non-eliminated arms at that time $\mathcal{A}_{t+1}$, (rather than over the entire set of arms $\mathcal{A}$, which would yield linear regret).
> There isn’t a particular motivation for the 0.01 value, in practice it is a tunable parameter and common magnitudes of uniform exploration are 0.01, 0.05 or 0.1. $\gamma$ can either be set as the magnitude of the uniform exploration over non-eliminated arms or as the lower-bound in a propensity of a non-eliminated arm. In our algorithm, it is the former so effectively the lower-bound of the propensity of a non-eliminated arm is $\gamma/K$ at worst. Given that it is a tunable parameter, in section B2 and Figure 3 of the supplemental material, we study the performance of DATS, TS-IPW and TS-DR for various levels of $\gamma$. That supplemental section shows that DATS is robust for various levels of $\gamma$ thanks to its variance-stabilizing adaptive weights whereas TS-IPW and TS-DR are a lot more sensitive to it so they benefit from tuning. That said, untuned DATS is still better than best-tuned TS-IPW and TS-DR.
> - That is an excellent point. We believe that best-arm identification algorithms, such as Top-Two Thompson sampling (TTTS), which is optimized for the sample complexity measure, could also benefit from adaptive inference as well since the usage of the sample averages is at the core of these algorithms as well. E.g., the propensity score of TTTS  is a function of the propensity score of TS sampling (Russo, 2016), so the benefits of DATS in the multi-armed bandit setting that we discussed in this paper are transferable and could motivate the DA-TTTS best-arm identification algorithm. Under similar arguments, correcting the bias of the sample averages used in TTTS with estimators such as IPW and DR will not be enough without the presence of stabilization weights that also have desirable variance properties as well and the normality guarantee. So, the application of this idea in the best-arm identification literature and the corresponding investigation under a sample complexity measure is an excellent source of further research, which we leave for our future work.
> - Thank you for the clarification question. A/B is implemented by setting the propensity score of every time period equal to $1/K$, where $K=6$ in this application, as in the actual A/B test that was exploring all arms uniformly at random for 10,000 test users. It can be found in lines 248-249 of the code submitted in the supplemental material.
> - Thank you for catching this--there is in fact a typo in the definition of $T_0$. The correct order of $T_0$ is $O(\log{T}/\Delta^2)$, and correspondingly the regret is still $\sqrt{T \log T}$. We have fixed this.
> - Per your suggestion, we have also substituted Corollary 1 with a single-line comment.
>
> Typos:
>
> Thank you for identifying the typos, this indicates close reading and we appreciate your help fixing them.

---

> > ### Comment · Reviewer_oqBT · 2021-08-19
> > **Response to the authors**
> >
> > I thank the authors for their detailed response to my review. While I am currently travelling (for the next 1.5 week) and in a very tight time schedule, I only give a high level response. I also read other reviews.
> >
> > I am happy that my suggestion lead to an improved dependency on the number of arms though it is still suboptimal. I disagree with the argument that it is not important to be optimal in K since in the applications you have in mind, the number of arms is small. Maybe other application could need larger values for K and if possible I think that it is better to at least try to be tight both in T and K.
> > Overall my opinion of the paper is unchanged and I keep my score of 6.

---

> > > ### Author Response · Authors · 2021-08-19
> > > **Re: Response to the authors**
> > >
> > > Dear Reviewer oqBT,
> > >
> > > Thank you very much for your response. We certainly agree with you that having a regret bound that is optimal also in K is desirable; our apologies for not being clear on that point. We just meant that in the applications we considered, K is small (so the difference wouldn't matter much in these), but as you pointed out, in general, and particularly for applications where there are a lot of arms, a sharper dependence on K is better. Having a regret bound that is optimal also in K to complement our regret bound that is optimal in T, the strong empirical advantage of our algorithm in several synthetic and real domains as well as the conceptual contribution of bringing asymptotically normal de-biasing inference in online bandits is the best possible outcome and making the regret bound tight in K is a direction of future work for us.
> > >
> > > We hoped that we could improve your already supportive opinion of our paper by using the author response period to further tighten our regret bound’s dependence on K by a factor of sqrt(K)  inspired by one of your review’s suggestions and also by highlighting in our response that in our paper we had worked hard to further investigate the dependence of our algorithm on K by simulating domains with a larger number of arms going up to K=50 and showing empirically that our algorithm has an optimal sqrt(K) dependence additionally to outperforming the baselines to complement our theory (Figure 1 in the submission).
> > >
> > > Thank you again for taking the time to review our work. We appreciate all your comments and suggestions that helped us improve our paper.

---

### Decision · Program_Chairs · 2021-09-27

**Decision:**

Accept (Poster)

**Comment:**

Three of the four reviewers recommended accepting the paper and I am happy to accept it. I encourage the authors to include in the final version the additional material that they mentioned in the rebuttal.